# Incentivizing LLM Reasoning via Reinforcement Learning with Functional Monte Carlo Tree Search

**Kongcheng Zhang**[1], **Qi Yao**[2], **Baisheng Lai**[2], **Jiaxing Huang**[3], **Wenkai Fang**[1],
**Dacheng Tao**[3], **Mingli Song**[1], **Shunyu Liu**[3*]
[1]Zhejiang University, [2]Alibaba Cloud Computing, [3]Generative AI Lab, College of Computing and Data Science, Nanyang Technological University, Singapore 639798
`zhangkc@zju.edu.cn, yq223369@alibaba-inc.com, shunyu.liu.cs@gmail.com`

## Abstract

In this work, we propose **R**einforced **F**unctional **T**oken **T**uning (RFTT), a novel reinforced fine-tuning framework that empowers Large Language Models (LLMs) with learn-to-reason capabilities. Unlike prior prompt-driven reasoning efforts, RFTT embeds a rich set of learnable functional tokens (*e.g.*, `<analyze>`, `<verify>`, `<refine>`) directly into the model vocabulary, enabling chain-of-thought construction with diverse human-like reasoning behaviors. Specifically, RFTT comprises two phases: (1) supervised fine-tuning performs prompt-driven tree search to obtain self-generated training data annotated with functional tokens, which warms up the model to learn these tokens for initial reasoning capability; and (2) online reinforcement learning further allows the model to explore diverse reasoning pathways through functional token sampling without relying on prompts, thereby facilitating effective self-improvement for functional reasoning. Extensive experiments demonstrate the superiority of the proposed RFTT on mathematical benchmarks and highlight its strong generalization capability to other general domains. Moreover, the performance of RFTT exhibits consistent gains with increased test-time computation through additional search rollouts. Our code and dataset are available at `https://github.com/sastpg/RFTT`.

## 1 Introduction

Recent advances in Large Language Models (LLMs), particularly exemplified by OpenAI-o1 (Jaech et al., 2024) and DeepSeek-R1 (Guo et al., 2025), have demonstrated sophisticated reasoning capabilities with remarkable success across various professional domains (Zhu et al., 2024; 2025; Tu et al., 2025), such as mathematical analysis (Hendrycks et al., 2021b; Cobbe et al., 2021; He et al., 2024; Zhang et al., 2025b) and algorithmic programming (Chen et al., 2021b; Austin et al., 2021; Zhuo et al., 2024). Despite the encouraging results achieved, learning to reason remains a crucial yet challenging task for LLMs as it necessitates high-quality reasoning data for training (Xu et al., 2025), especially for smaller LLMs with 7B or 8B parameters (Guan et al., 2025). Thus, previous attempts often rely on stronger models or human annotators to generate high-quality Chain-of-Thought (CoT) data (Min et al., 2024b; Lightman et al., 2024; Huang et al., 2024; Wang et al., 2024a); however, such approaches inevitably incur substantial costs (Wang et al., 2024b; Guan et al., 2025; Zhang et al., 2025c) and are prone to limited scalability (Li et al., 2023; Ahn et al., 2024).

To address these limitations, existing studies explore self-improvement mechanisms where LLMs can iteratively refine their reasoning capabilities through self-generated rationales without human-curated training data (Hao et al., 2023; Zhang et al., 2024a; Wu et al., 2024; Zhang et al., 2024b; Yao et al., 2024; Fang et al., 2025). For instance, early techniques such as CoT prompting enable models to generate multi-step reasoning paths for unlabeled questions, followed by rejection sampling to select high-confidence solutions as training data (Yuan et al., 2023; Liu et al., 2024c; Brown et al., 2024). More recently, DeepSeek-R1 (Guo et al., 2025) and its follow-up works (Zeng et al., 2025; Xie et al.,

---

*Corresponding author.

2025; Yu et al., 2025; Hu et al., 2025; Zhang et al., 2025a) have demonstrated that Reinforcement Learning (RL) offers a powerful paradigm for autonomously incentivizing LLMs with advanced reasoning capabilities, leading to the spontaneous emergence of complex "aha moment" behaviors.

However, some studies (Shah et al., 2025; Yue et al., 2025) have raised critical perspectives, suggesting that RL may not introduce fundamentally new reasoning capabilities, but rather increase the likelihood of sampling correct outputs that the initial model already knows. Moreover, RL enhances sampling efficiency at the cost of reducing reasoning diversity, potentially limiting the exploration capability of LLMs. This reveals a fundamental dilemma: effective learning to reason necessitates both a strong initial reasoning capability and sustained exploratory diversity. Without sufficient initial competence, the model may struggle to generate informative feedback during self-play training. Furthermore, a lack of exploratory diversity can cause the model to converge prematurely on suboptimal reasoning patterns, hindering the discovery of more optimal reasoning paths.

In this work, we propose *Reinforced Functional Token Tuning* (RFTT), a novel reinforced fine-tuning framework that introduces learnable functional tokens (*e.g.*, `<analyze>`, `<verify>`, `<refine>`) into the model vocabulary to facilitate self-training learn-to-reason. Technically, RFTT operates in two phases: (1) During the Supervised Fine-Tuning (SFT) warmup phase, RFTT employs functional prompt-guided Monte Carlo Tree Search (MCTS) to construct reasoning trees, where both correct and incorrect solution paths are merged to synthesize human-like reasoning paths. These self-generated paths explicitly connect reasoning nodes through functional tokens, thereby forming structured training data for learning to reason with these tokens, which equips the model with **initial reasoning capability**. (2) In the online RL phase, the model transitions from prompt-guided to token-guided reasoning by sampling functional tokens to autonomously expand various reasoning nodes. Through **diverse exploration of reasoning trees**, the model reinforces high-value reasoning paths, ultimately achieving autonomous self-improvement of functional reasoning capabilities.

Our core contributions are summarized as follows:

- We propose a new learn-to-reason paradigm that pioneers the integration of learnable functional tokens into the model vocabulary, enabling the model to establish internalized token-guided reasoning patterns rather than relying on external prompt-guided constraints.

- We devise RFTT, a novel reinforced fine-tuning framework for learning to reason. The SFT phase bootstraps initial reasoning through *self-generated* training data annotated with functional tokens, while the RL phase enables diverse exploration of reasoning paths through token-guided tree traversal, achieving *self-improvement* for functional reasoning.

- Extensive experiments conducted on various mathematical benchmarks demonstrate that RFTT yields significantly superior results to state-of-the-art counterparts. Notably, the performance of RFTT continues to improve as the number of search rollouts increases during inference.

## 2  RELATED WORKS

**Learn-to-Reason.** The "learn-to-reason" paradigm aims to endow LLMs with complex reasoning capabilities through dedicated training. Due to the scarcity of stepwise annotated reasoning trajectories, previous methods have primarily focused on two strategies: synthesizing long-CoT (Wei et al., 2022; Kojima et al., 2022) trajectories for supervised fine-tuning (Trung et al., 2024; Zhang et al., 2024a; Guan et al., 2025; Min et al., 2024a), and generating reasoning preference pairs for preference optimization (Rafailov et al., 2023; Lai et al., 2024; Liu et al., 2025). More recently, DeepSeek-R1 (Guo et al., 2025) demonstrated that reinforcement learning with verifiable rewards (Lambert et al., 2024) (RLVR) can elicit advanced reasoning abilities without curated trajectories, similar to traditional reinforcement learning (Jiang et al., 2021; 2022; Liu et al., 2023; 2024b). However, subsequent studies (Shah et al., 2025; Yue et al., 2025) indicate that RLVR may degrade the exploration capability and reduce the reasoning diversity of LLMs. In RFTT, we mitigate this issue by employing functional tree search to enhance the exploration of the solution space.

**Tree-Search Reasoning.** While most LLMs have typically adopted an auto-regressive reasoning manner, there has been a trend to engage in more complicated reasoning architectures like trees (Wang et al., 2023b;a; Koh et al., 2024; Zhang et al., 2024c; Ding et al., 2025). Recently, various methods for exploring tree structures have been devised to identify optimal reasoning trajectories, *e.g.*, tree of thought (Yao et al., 2023), graph of thoughts (Besta et al., 2024), and Monte Carlo tree search (Hao

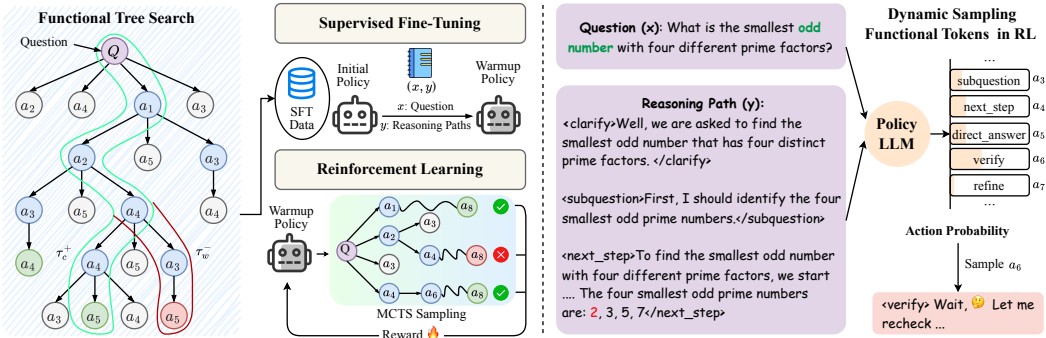

Figure 1: A conceptual illustration of reasoning path generation based on functional tree search and our training framework. RFTT comprises two phases: supervised fine-tuning warmups the model with initial reasoning capability by functional token-annotated data, while online reinforcement learning allows the model to directly sample functional tokens from its vocabulary to autonomously expand reasoning trees for diverse exploration.

et al., 2023; Chen et al., 2024; Zhang et al., 2024a). Furthermore, several studies (Shen et al., 2025; Hou et al., 2025) also incorporate tree search for RL training. However, these direct tree-search methods struggle with limited exploration due to the homogeneous reasoning paths and the large search space. Although rStar (Qi et al., 2024) utilizes functional prompts in tree search to diversify and constrain the search process, they only act as external constraints in the inference phase while we introduce learnable functional tokens during model training to facilitate effective and efficient exploration, thus improving the model's intrinsic reasoning ability.

## 3 METHODOLOGY

In this section, we first provide the problem formulation. Then we further detail the proposed two phase learn-to-reason framework of RFTT. The whole procedure is outlined in Figure 1.

### 3.1 PROBLEM FORMULATION

We consider a complex reasoning task as a multi-step reasoning generation problem, which decomposes the problem into a sequence of simpler intermediate steps through structured reasoning operations. Assuming a problem $x$ requires $T$ intermediate reasoning steps, the reasoning path can be denoted as $\tau = \{s_0, s_1, s_2, ..., s_T\}$, where $s_0 = x$ and $s_t$ is the $t$-th intermediate reasoning steps. Furthermore, we model this process as a tree search, where the root node represents the problem $x$, the edges denote the action space $\mathcal{A}$ (which can be special tokens or prompts), and children nodes are next steps generated by a policy model $\pi_\theta$ with parameter $\theta$ under corresponding actions. Each distinct path from the root to the terminal node denotes a sequence of decision-making steps that constitutes a potential solution trajectory $\tau$. For any given problem $x$, the subset of its solution space $\mathcal{T} = \{\tau_1, \tau_2, ..., \tau_N\}$ can be extracted from the tree structure. Our aim is to sample diverse reasoning paths and ultimately derive optimal solutions.

Considering the problem setup above, we can transfer the task of better reasoning to better selecting actions based on previous steps. In this work, we focus on how to inspire intrinsic reasoning abilities with functional tree search. The conventional natural language generation task often regards vocabulary-level token sampling as actions. If we adopt vocabulary-level actions to conduct tree search on LLMs, the action space can be extremely large, leading to a sharp increase in computational complexity and reduced search efficiency. Thus, we design a series of human-like reasoning behaviors as the action space of the LLM policy. During the preliminary warmup phase for SFT, we employ functional prompts as the action space to induce the model for tree node generation with different semantics, while transitioning to functional tokens for exploratory tree search during RL optimization.

### 3.2 FUNCTIONAL MONTE CARLO TREE SEARCH FOR DATA GENERATION

The trajectories generated by direct tree search methods are often homogeneous due to the inherent preference for syntactic patterns in LLMs. To enrich LLMs with sophisticated reasoning capabili-

ties, we first analyze the reasoning behaviors of leading LRMs. Inspired by recent work (Gandhi et al., 2025; Luo et al., 2025), we categorize several functional tokens as actions to emulate how humans engage in complex reasoning: `<clarify>` ($a_1$), `<analysis>` ($a_2$), `<subquestion>` ($a_3$), `<next_step>` ($a_4$), `<direct_answer>` ($a_5$), `<verify>` ($a_6$), `<refine>` ($a_7$) and `<output>` ($a_8$). We then introduce MCTS guided by functional prompts as actions before the SFT phase to obtain diverse reasoning nodes. Through iterative MCTS rollouts, we can extract multiple reasoning trajectories $\mathcal{T} = \{\tau_1, \tau_2, ..., \tau_N\}$ for each question from a tree structure. More details of MCTS procedures and functional token descriptions with corresponding functional prompts are presented in Appendix D and Appendix E, respectively. The following steps describe how we utilize these trajectories to construct structured human-like reasoning paths for SFT.

**Cross Verification.** To construct a complex reasoning structure that incorporates self-verification for SFT, we generate an extra node via cross verification between two trajectories and thereby seamlessly concatenate them. Given a question $x$ and its ground truth $y$, we can obtain a set of candidate reasoning trajectories $\mathcal{T}$ as stated before. We first choose a correct trajectory with the highest average process rewards given by a process reward model $\tau_c = \arg\max_{\tau_i \in \mathcal{T}_c} \bar{R}(\tau_i)$, where $\mathcal{T}_c$ is the set of correct trajectories and $\bar{R}(\tau_i)$ represents the average rewards of all intermediate steps in trajectory $\tau_i$. We then identify

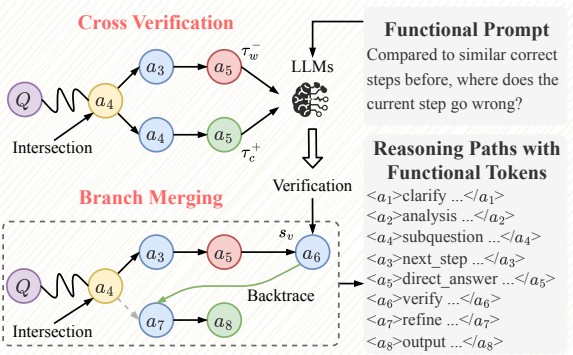

Figure 2: An illustration of cross verification and branch merging for SFT data construction.

a subset $\mathcal{T}_w \subset \mathcal{T}$ consisting of incorrect trajectories that intersect with $\tau_c$. From $\mathcal{T}_w$, we extract the reasoning path with the lowest average reward: $\tau_w = \arg\min_{\tau_i \in \mathcal{T}_w} \bar{R}(\tau_i)$. If $\tau_w$ is non-empty, we define the overlapping nodes between $\tau_c$ and $\tau_w$ as $\tau^+ = \tau_c \cap \tau_w$, and the distinct nodes of them as $\tau_c^+ = \tau_c - \tau^+$ and $\tau_w^- = \tau_w - \tau^+$. Through this approach, we can yield a node $s_v \sim \pi_\theta (\cdot \mid \tau_c \cup \tau_w^- \cup \{p\})$ that verify why an error occurs in $\tau_w^-$ using a prompt $p$ = "Compared to similar correct steps before, where does the current step go wrong?".

**Branch Merging.** We adopt branch merging to form structured reasoning paths that feature self-verification and self-correction: $\tau_f = \tau^+ \cup \tau_w^- \cup \{s_v\} \cup \tau_c^+$. Finally, we concatenate intermediate reasoning steps in trajectories by functional tokens, generating a trainable reasoning path $\mathcal{D}_{\text{SFT}} = \left\{ \otimes_{s_i \in \tau_f} (<a_i>s_i</a_i>) \right\}$ for supervised fine-tuning, where $\otimes$ denotes the string concatenation operation, $s_i$ is the intermediate reasoning step, and $a_i$ is the corresponding functional token.

## 3.3 REINFORCED FUNCTIONAL TOKEN TUNING

Our primary goal is to empower LLMs with self-training learn-to-reason capabilities. To achieve this, we introduce a two-phase training strategy: the SFT phase learns functional tokens for **initial reasoning capability** by self-generated training data annotated with these tokens, while the RL phase further enables autonomous **diverse exploration** of reasoning paths through functional token-guided tree search, thereby reinforcing high-value reasoning paths.

### 3.3.1 SUPERVISED FINE-TUNING

In this stage, an initial base model $\pi_\theta$ is warmuped up on the dataset $\mathcal{D}_{\text{SFT}}$ that consists of the tuples of "(question, reasoning path)" to obtain the SFT model $\pi_{\text{SFT}}$. Specifically, we treat functional tokens as new special tokens that can be embedded directly into the model vocabulary. During the SFT phase, the model learns to reason with these functional tokens without relying on prompts.

### 3.3.2 REINFORCEMENT LEARNING WITH FUNCTIONAL TREE SEARCH

After being trained on the data with functional tokens, the policy model $\pi_\theta$ can explicitly and dynamically select functional tokens and further generate the specific reasoning steps associated with these tokens. This enables the transition from prompt-guided to token-guided reasoning just by sampling functional tokens. Through this design, the policy model can reinforce the use of key

functional token combinations via a form of online tree search, significantly improving its learning efficiency. Specifically, the policy model learns by iteratively searching, evaluating the process in each rollout, and updating its parameters, as illustrated in Figure 1.

Like AlphaZero (Silver et al., 2018), the functional token is sampled based on the log-likelihood scores if there exist unexplored actions, otherwise select them according to the UCT score:

$$a_t = \begin{cases} \arg\max_{a \in U(s_t)} \pi_\theta(a|s_{0:t}), & U(s_t) \neq \emptyset, \\ \arg\max_{a \in \mathcal{A}(s_t)} \mathtt{UCT}(s_t, a), & \text{otherwise}, \end{cases}, \quad \mathtt{UCT}(s_t, a) = \frac{Q(s_t, a)}{N(s_t, a)} + c \cdot \sqrt{\frac{\ln N(s_t)}{N(s_t, a)}}, \quad (1)$$

where $U(s_t)$ denotes unexplored functional tokens of intermediate reasoning step $s_t$, and $s_{0:t}$ represents all intermediate steps from timestep $0$ to $t$. In UCT score, $N(s_t)$ denotes the number of times step $s_t$ has been visited in previous iterations, $N(s_t, a)$ denotes the number of times action $a$ has been selected after step $s_t$, $Q(s_t, a)$ is the cumulative reward value of selecting action $a$ after step $s_t$ obtained from previous simulations, and $c$ is a constant that balances exploitation and exploration.

Furthermore, the reward function based on the rule-verified answer extractor $\mathtt{ANS}(\cdot)$ is defined as:

$$R_t(s_{0:t}, a_t, s_{t+1}) = \mathtt{RM}(s_{0:t}, a_t, s_{t+1}) - \beta \cdot \mathtt{KL}(t), \quad (2)$$

$$\mathtt{RM}(s_{0:t}, a_t, s_{t+1}) = \begin{cases} 1, & \mathtt{ANS}(s_{t+1}) = y, \\ 0.1, & \mathtt{ANS}(s_{t+1}) \neq \mathtt{null}, \neq y, \\ \sigma, & \mathtt{ANS}(s_{t+1}) = \mathtt{null}, \end{cases} \quad \mathtt{KL}(t) = \log\left(\frac{\pi_\theta(a_t|s_{0:t})}{\pi_{\mathrm{SFT}}(a_t|s_{0:t})}\right), \quad (3)$$

where the SFT model $\pi_{\mathrm{SFT}}$ serves as the reference model, $\beta$ is KL coefficient, and $\sigma$ is the process reward. We can use the Process Reward Model (PRM) to obtain the value of $\sigma$. Note that $\sigma$ will be assigned a value of $0$ if only outcome rewards are used.

During online RL, the model parameters are optimized using the Reinforce++ algorithm (Hu, 2025). The core policy objective combines clipped updates with normalized advantage values through the following loss function:

$$\mathcal{L}_{RL}(\theta) = -\mathbb{E}_t\left[\min\left(\frac{\pi_\theta(a_t|s_{0:t})}{\pi_{\theta_{\mathrm{old}}}(a_t|s_{0:t})}\hat{A}_t, \mathrm{clip}\left(\frac{\pi_\theta(a_t|s_{0:t})}{\pi_{\theta_{\mathrm{old}}}(a_t|s_{0:t})}, 1-\epsilon, 1+\epsilon\right)\hat{A}_t\right)\right], \quad (4)$$

where $\pi_{\theta_{\mathrm{old}}}$ is the previous policy model, $\epsilon$ is the clipping coefficient, and the normalized advantage value $\hat{A}_t$ is calculated based on the reward $R_t$.

Overall, the transition from prompt-guided to token-guided reasoning enables the effective and efficient exploration of the reasoning paths during the RL phase. The policy model learns to reinforce high-value reasoning paths through functional token-guided tree search and achieves self-improvement for functional reasoning.

## 3.4 EFFECTIVENESS ANALYSIS

We briefly provide insights into why RFTT can outperform current reinforced fine-tuning methods.

- **High entropy exploration.** Diverging from independent rollouts like GRPO, our method implements a tree-structured branching strategy at each node guided by the probability of functional tokens. A low-entropy node over functional tokens (probability concentrates on one action) suppresses branching because alternatives are rarely selected. Therefore, this mechanism adaptively allocates exploration resources to paths in the reasoning space that have **increasing entropy**, thus promoting a broader exploration of diverse reasoning patterns.

- **Distinct advantage attribution.** Our functional tree search sampling produces a large number of trajectories with **shared tokens but divergent branches**. Through implicit advantage attribution, the RL algorithm enables the model to effectively internalize the distinctions among different reasoning behaviors. Since shared tokens receive the same aggregated advantage signals from the entire group, while branching tokens obtain distinct advantages specific to their exploration paths, they contribute differently to the final optimization objective.

Consequently, the optimization process improves exploration efficiency while facilitating precise learning at critical reasoning steps in different states.

# 4 EXPERIMENTS

To demonstrate the effectiveness of the proposed RFTT method, we conduct experiments on multiple mathematical and general reasoning benchmarks. Our evaluation seeks to answer the following questions: (1) Does RFTT outperform existing reinforcement fine-tuning baselines? (Section 4.3 and Appendix B) (2) How do the configurations of functional tokens and different training/rollout strategies affect the performance? (Section 4.4) (3) Can RFTT expand the exploration boundaries to a higher capacity ceiling than the initial model? (Section 4.5)

## 4.1 EXPERIMENTAL SETUPS

**Datasets.** For training, we only adopt the training set of the MATH (Hendrycks et al., 2021b) dataset as our training data. For evaluation, we employ five established mathematical reasoning benchmarks: MATH-500 (Lightman et al., 2024), GSM8K (Cobbe et al., 2021), SVAMP (Patel et al., 2021), Olympiad Bench (He et al., 2024), and AMC[1]. Moreover, we evaluate on general reasoning datasets like MMLU-Pro (Wang et al., 2024c), GPQA (Rein et al., 2024), CommonsenseQA (Talmor et al., 2019), FOLIO (Han et al., 2024), TableBench (Wu et al., 2025), and CRUXEval (Gu et al., 2024) to validate the generalization of RFTT. To ensure evaluation efficiency and prevent data contamination, we use the MATH-500 for MATH evaluations while retaining full test sets for other datasets. More information about our training and evaluation dataset is listed in Appendix C.

**Base Models.** RFTT is generally applicable to a wide range of LLMs. In our experiments, we select LLaMA-3.1-8B-Instruct (Dubey et al., 2024), Qwen-2.5-7B-Instruct (Yang et al., 2024b), and Qwen-3-4B-Base (Yang et al., 2025) as backbones. By focusing on LLMs with relatively small parameters under 10B, we expect that RFTT enables the smaller model to learn to reason via self-play, ultimately achieving results comparable to or exceeding larger LLMs in complex reasoning tasks.

**Baselines.** We compared RFTT against four categories of baselines in our experiments: (1) *Superior LLMs*, including leading closed-source models and open-source models; (2) *Model fine-tuning methods*, including SFT, ReFT (Trung et al., 2024), MCTS-DPO (Xie et al., 2024), TreeRL (Hou et al., 2025) and GRPO (Shao et al., 2024) with Deepseek-R1 prompt template; (3) *In-context learning methods*, including Few-shot CoT and CoT+SC@4. (4) *Tree search methods*, including ResT-MCTS* (Zhang et al., 2024a), rStar (Qi et al., 2024) and LLaMA-Berry (Zhang et al., 2024c).

**Evaluation Metric.** We employ Pass@1 accuracy across all benchmark datasets as our evaluation metric, with correctness determined through comparison between generated outputs and ground truth. To extract answers reliably, we require the model to wrap its final answer in `boxed{}`.

## 4.2 IMPLEMENTATION DETAILS

All experiments run on $8\times$A800-80GB GPUs. While performing MCTS to collect diverse reasoning paths via various functional prompts, we use Qwen-2.5-7B-Instruct as the generator and math-shepherd-mistral-7b-prm (Wang et al., 2024b) as the process reward model. Note that for the Qwen experiment, the model was trained on its own generated data. For convenience, we used this data directly in other experiments without regenerating it. We present the performance results after both SFT and RL, and the findings are consistent. We set the maximum search depth to 15 and perform 16 rollouts per question. For sampling each step, we employ the vLLM engine (Kwon et al., 2023) to accelerate with the temperature set to 0.9, top p set to 0.8, and max tokens set to 1024. We spend approximately one day searching through 1.2k questions using 64 concurrent processes, and then gather 1k SFT data as described in Section 3.2.

We use LLaMA-Factory (Zheng et al., 2024) and OpenRLHF (Hu et al., 2024) for SFT and RL, respectively. During SFT, the training batch size is 128, the learning rate is 7e-6, and the cutoff length is 8192. We train the initial model for 10 epochs on about 1k CoT data with functional tokens and select the checkpoint with the best performance as the initial policy model for RL. During RL, we implement functional token-guided MCTS to sample different reasoning paths for a question. After that, we can choose to score the intermediate reasoning nodes in the tree. At each training step, the model searches for 16 reasoning paths per question with a batch of 16 distinct questions. The learning rate for the policy model is 5e-7, the temperature is 0.95, and the KL coefficient is 0.01.

---

[1]https://huggingface.co/datasets/AI-MO/aimo-validation-amc

Table 1: Accuracy of our proposed RFTT and baselines across different mathematical reasoning benchmarks. The best results in each box are highlighted in **bold**. The proposed RFTT significantly boosts the performance of smaller LLMs across all datasets.

| Model | Setting | In-Domain | Close-Domain | | | | AVERAGE |
|---|---|---|---|---|---|---|---|
| | | MATH-500 | GSM8K | SVAMP | Olympiad Bench | AMC | |
| *Superior LLMs* | | | | | | | |
| GPT-4o | Closed-source | 76.6 | **96.1** | **93.8** | **43.3** | 47.5 | 71.46 |
| GPT-4o mini | Closed-source | 70.2 | 93.2 | 89.2 | 35.8 | 55.0 | 68.68 |
| o1-preview | Closed-source | **85.8** | 93.0 | - | - | **90.0** | - |
| Qwen-2.5-14B-Instruct | Open-source 14B | 80.0 | 93.2 | 91.6 | 39.4 | 52.5 | 71.34 |
| *Small LLMs* | | | | | | | |
| Qwen-3-4B-Base | Zero-shot CoT | 58.6 | 80.6 | 78.9 | 31.4 | 50.0 | 59.90 ↑0.00 |
| | Few-shot CoT | 64.8 | 87.8 | 80.6 | 35.3 | 57.5 | 65.20 ↑5.30 |
| | CoT+SC@4 | 67.0 | 87.6 | 81.2 | 35.8 | 52.5 | 64.82 ↑4.92 |
| | ReFT | 79.2 | 94.4 | 91.7 | 45.8 | 70.0 | 76.22 ↑16.32 |
| | GRPO | 75.6 | 92.7 | 88.4 | 42.9 | 67.5 | 73.42 ↑13.52 |
| | MCTS-DPO | 73.8 | 90.2 | 86.9 | 39.7 | 60.0 | 70.12 ↑10.22 |
| | TreeRL | 82.2 | 95.3 | 92.8 | 46.5 | 70.0 | 77.36 ↑17.46 |
| | **SFT Warmup (Ours)** | 68.2 | 89.5 | 85.6 | 38.9 | 62.5 | 68.94 ↑9.04 |
| | **RFTT (Ours)** | **83.4** | **96.1** | **94.7** | **48.1** | **75.0** | **79.46 ↑19.56** |
| Qwen-2.5-7B-Instruct | Zero-shot CoT | 72.0 | 91.1 | 88.5 | 35.1 | 45.0 | 66.34 ↑0.00 |
| | Few-shot CoT | 75.6 | 91.6 | 90.2 | 36.5 | 52.5 | 69.28 ↑2.94 |
| | CoT+SC@4 | 76.4 | 92.1 | 90.6 | 38.2 | 50.0 | 69.46 ↑3.12 |
| | ReFT | 75.8 | 94.1 | 89.3 | 38.1 | 60.0 | 71.46 ↑5.12 |
| | GRPO | 76.2 | 93.7 | 90.6 | 36.3 | 55.0 | 70.36 ↑4.02 |
| | MCTS-DPO | 74.6 | 92.5 | 89.8 | 37.2 | 55.0 | 69.82 ↑3.48 |
| | TreeRL | 78.4 | 94.8 | 92.3 | 39.6 | 62.5 | 73.52 ↑7.18 |
| | **SFT Warmup (Ours)** | 73.2 | 91.9 | 87.4 | 36.8 | 60.0 | 69.86 ↑3.52 |
| | **RFTT (Ours)** | **79.8** | **95.2** | **93.0** | **40.3** | **70.0** | **75.66 ↑9.32** |
| LLaMA-3.1-8B-Instruct | Zero-shot CoT | 50.6 | 84.5 | 78.2 | 18.6 | 17.5 | 49.88 ↑0.00 |
| | Few-shot CoT | 53.0 | 85.1 | 82.0 | 20.0 | 20.0 | 52.20 ↑2.32 |
| | CoT+SC@4 | 52.2 | 86.0 | 85.6 | 19.7 | 25.0 | 53.70 ↑2.82 |
| | ReFT | 55.0 | 90.8 | 84.3 | 24.9 | 42.5 | 59.50 ↑9.62 |
| | GRPO | 54.0 | 88.2 | 83.4 | 24.0 | 22.5 | 54.42 ↑4.54 |
| | MCTS-DPO | 53.6 | 87.4 | 82.7 | 23.1 | 25.0 | 54.36 ↑4.48 |
| | TreeRL | 57.8 | **92.6** | 86.1 | 26.7 | 47.5 | 62.14 ↑12.26 |
| | **SFT Warmup (Ours)** | 53.2 | 86.4 | 82.9 | 24.6 | 40.0 | 57.40 ↑7.52 |
| | **RFTT (Ours)** | **60.2** | 91.9 | **87.5** | **29.8** | **55.0** | **64.88 ↑15.0** |

Table 2: Performance of RFTT on out-of-domain benchmarks. Despite being trained only on math datasets, RFTT exhibits strong transferability.

| Model | Setting | MMLU-Pro | GPQA | CommonsenseQA | FOLIO | TableBench | CRUXEval |
|---|---|---|---|---|---|---|---|
| Qwen-2.5-7B-instruct | Few-shot CoT | 55.6 | **36.4** | 81.6 | 72.9 | 43.6 | **58.3** |
| | **RFTT (Ours)** | **57.2** | 35.6 | **82.8** | **73.9** | **44.4** | 57.5 |
| LLaMA-3.1-8B-instruct | Few-shot CoT | 48.3 | 32.8 | 76.9 | 64.5 | 32.8 | 38.6 |
| | **RFTT (Ours)** | **50.8** | **33.4** | **78.1** | **65.6** | **34.2** | **39.5** |

## 4.3 MAIN RESULTS

**Results on Different Reasoning Benchmarks.** We conduct a comprehensive evaluation of the effectiveness of RFTT across different mathematical benchmarks. Table 1 presents a comparative analysis between our framework and state-of-the-art baselines. We highlight three key findings: (1) RFTT can significantly improve the complex reasoning proficiency of small-parameter LLMs. For example, LLaMA-3.1-8B-Instruct initially achieves 50.6% accuracy on the MATH-500 benchmark using the CoT prompting technique. However, after training by RFTT, its Pass@1 accuracy improves to 60.2%, surpassing self-consistency sampling (CoT+SC@4). Similarly, Qwen-2.5-7B-Instruct with RFTT achieves performance on par with Qwen-2.5-14B-Instruct (79.6% vs. 80% on MATH-500), indicating that smaller models have developed robust reasoning capabilities without any reasoning prompts. (2) RFTT outperforms ReFT on various mathematical reasoning tasks. We observe that RFTT achieves consistent performance improvements over ReFT, notably surpassing it by an average margin of 5% across different benchmark evaluations. This result demonstrates the robustness of RFTT and the enormous potential for further self-training. (3) RFTT has also shown strong generalization ability on other challenging reasoning problems, including AMC, Olympiad Bench,

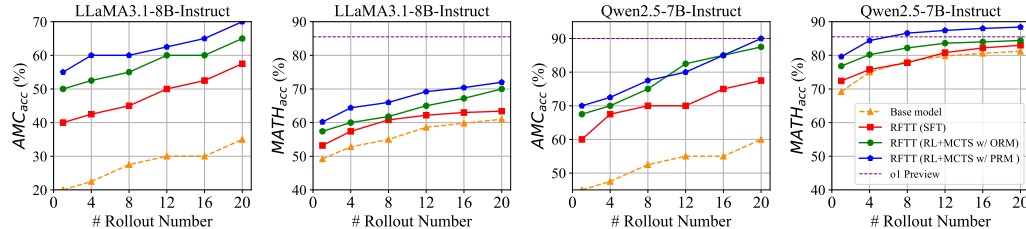

Figure 3: Performance gains under scaling up the inference-time computation.

GPQA, and MMLU-Pro. As mentioned in Section 4.1, our training dataset only consists of the training data from MATH, and there may be a risk of over-optimization on well-known testing benchmarks such as MATH-500 and GSM8K. Our results in Table 2 reveal that RFTT remains effective even when applied to unseen general problem sets. More results can be found in Appendix B.

**Scaling Inference-Time Computation.** We integrate learnable functional tokens into the model vocabulary in RFTT, enabling the model to establish internalized token-guided tree search. We use MCTS to augment the policy model trained by RFTT without any guidance of reward models. Specifically, we use the likelihood (confidence) of majority voting as the reward value in the MCTS simulation stage. Figure 3 shows the performance of scaling inference-time computation by comparing the accuracy of policy models with different training strategies across different rollout numbers. The accuracy of one rollout corresponds to the Pass@1 accuracy of the policy model. We highlight two key observations: (1) Scaling inference-time computation enhances mathematical reasoning across all benchmarks, albeit with distinct trends observed. On MATH-500, the policy models demonstrate a slow improvement after 16 rollouts, whereas on AMC, the accuracies continue to improve steadily. (2) With 8 and 20 rollouts on MATH-500 and AMC benchmarks, respectively, the performance of the trained policy model exceeds o1-preview, demonstrating its effectiveness.

**Tree Search Comparison.** Our proposed RFTT can autonomously select the next high-value functional token based on a partially generated reasoning trajectory, thus enabling a more efficient exploration of the reasoning space without frequently interrupting the generation process. Here, we compare the accuracy and search efficiency of RFTT with other tree-based reasoning methods on LLaMA-3.1-8B-Instruct.

Table 3: Comparison of different tree search methods.

| Dataset | Method | Accuracy | Time per Question |
|---------|--------|----------|-------------------|
| GSM8K | rStar | 92.1% | 276 s |
| | LLaMA-Berry | 94.9% | 339 s |
| | **Ours** | **95.2%** | **81** s |
| MATH-500 | rStar | 61.0% | 526 s |
| | LLaMA-Berry | 69.4% | 674 s |
| | **Ours** | **72.0%** | **131** s |

Specifically, we measure the average time needed for a single question and the proportion of correct answers under 16 rollouts across datasets. As shown in Table 3, our framework achieves higher accuracy while substantially reducing time complexity compared to other methods. It outperforms rStar, which relies on extensive node expansion to select a potentially better action, and LLaMA-Berry, which generates a complete solution instead of a single step for each node.

## 4.4 ABLATION STUDIES

We first investigate our design of functional tokens. To further demonstrate the advantage of our method, we ablate core components of RFTT: (1) *RFTT w/o SFT Warmup* directly trains the original model using rule-based RL; (2) *RFTT w/o MCTS* employs random sampling and rule-based RL after SFT Warmup; (3) *RFTT w/o PRM* employs MCTS sampling and rule-based RL after SFT Warmup.

**The necessity and generalization of functional tokens.** We conduct an ablation study on MATH-500 using Qwen-2.5-7B-Instruct trained after RFTT by masking the possibility of a certain functional token. Results in Table 4 highlight the complementary effectiveness of all functional tokens. Notably, tokens like `<verify>` and `<refine>` have a significant contribution, and their absence leads to performance degradation. Moreover, we generate responses on MATH-500 and prompt GPT-4o to classify each reasoning step by type. Figure 4 shows that the semantics of functional tokens align strongly with their intended behaviors. This supports our design of introducing eight human-inspired functional tokens, which serve as a compact yet expressive action space for reasoning exploration.

Table 6: Ablation study on different components of RFTT.

| Model | Setting | MATH-500 | GSM8K | SVAMP | Olympiad Bench | AMC | AVERAGE |
|---|---|---|---|---|---|---|---|
| Qwen-2.5-7B-Instruct | RFTT w/o SFT Warmup | 74.8 | 91.2 | 90.6 | 37.9 | 52.5 | 69.40 |
| | RFTT w/o MCTS | 75.2 | 93.8 | 91.2 | 39.0 | 62.5 | 72.34 |
| | RFTT w/o PRM | 77.2 | 94.1 | 92.4 | 38.4 | 67.5 | 73.92 |
| | **RFTT (Ours)** | **79.8** | **95.2** | **93.0** | **40.3** | **70.0** | **75.66** |
| LLaMA-3.1-8B-Instruct | RFTT w/o SFT Warmup | 54.8 | 84.9 | 83.7 | 24.0 | 17.5 | 53.00 |
| | RFTT w/o MCTS | 56.2 | 91.5 | 84.0 | 29.1 | 42.5 | 60.66 |
| | RFTT w/o PRM | 57.4 | **92.2** | 86.3 | 28.6 | 50.0 | 62.90 |
| | **RFTT (Ours)** | **60.2** | 91.9 | **87.5** | 29.8 | **55.0** | **64.88** |

We additionally perform an analysis using Deepseek-R1 on 1,000 randomly selected questions from MMLU-Pro (a dataset that spans STEM, social sciences, law, and health). The sampled reasoning trajectories are broken down into discrete steps using predefined rules (*e.g.*, newline delimiters). We then employ GPT-4o to assess whether each step could be mapped to one of our functional tokens. As presented in Table 5, approximately 98.4% of the steps aligned with the intended semantic coverage of our token set, which demonstrates *the generation of our functional tokens across diverse tasks*.

Table 4: Ablation study on masking different functional tokens.

| Functional Tokens | Accuracy |
|---|---|
| w/o $a_1$ | 79.0% |
| w/o $a_2$ | 77.6% |
| w/o $a_3$ | 79.4% |
| w/o $a_4$ | 73.4% |
| w/o $a_5$ | 75.0% |
| w/o $a_6$ | 72.8% |
| w/o $a_7$ | 72.6% |
| All (Ours) | **79.8%** |

Table 5: Coverage rate of functional tokens on general domains.

| Functional Tokens | Coverage Rate |
|---|---|
| token $a_1$ | 8.2% |
| token $a_2$ | 13.9% |
| token $a_3$ | 4.3% |
| token $a_4$ | 28.7% |
| token $a_5$ | 14.5% |
| token $a_6$ | 13.8% |
| token $a_7$ | 12.3% |
| token $a_8$ | 2.7% |

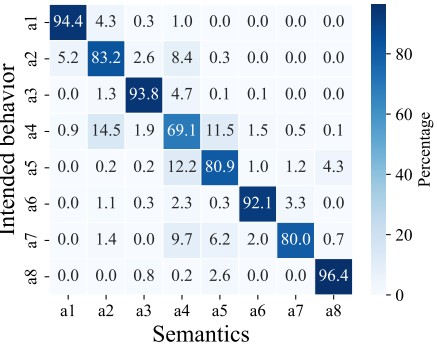

Figure 4: The relationship between semantics and behaviors of functional tokens.

**The Effectiveness of SFT Warmup.** The SFT phase embeds the functional tokens into the model vocabulary, enabling self-improvement with token-guided exploration during the RL phase. To verify the effectiveness of our SFT phase, we conduct experiments to directly train the original model without SFT warmup using pure RL. As compared in the first two lines of Table 6, even with the same random sampling method, the model warmuped with SFT (RFTT w/o MCTS) surpasses the pure RL model (RFTT w/o SFT warmup) by a large margin (2.9% for Qwen-2.5-7B-Instruct and 7.7% for LLaMA-3.1-8B-Instruct), indicating that the token-guided reasoning can explore the reasoning space much more effectively after SFT initialization.

**The Effectiveness of MCTS.** Our functional token-driven MCTS enables a more effective exploration of the solution space in both inference and training. We conducted an ablation study on MCTS sampling during the RL phase to investigate the influence of data diversity, as shown in Table 6. The outcomes reveal that random sampling in RL yields lower performance compared to sampling with MCTS, thereby underscoring the significance of data diversity in learning a better policy. The results show a 1.5% performance gap between random sampling (mean 72.34) and MCTS-enhanced sampling (mean 73.92) after rule-based RL across 5 benchmarks. Thus, we conclude that functional token-driven MCTS can explore diverse reasoning paths, providing a superior foundation for RL.

**The Effectiveness of PRM.** The results in Table 6 demonstrate that the use of PRM in RFTT consistently outperforms that of rule-based RL (RFTT w/o PRM) by about 2% on both LLaMA and Qwen on average. This might benefit from fine-grained rewards for intermediate reasoning steps, which mitigate error accumulation in long reasoning paths. Therefore, the integration of PRM is valuable for reasoning where rule-based rewards fail to capture intermediate reasoning quality.

## 4.5 DISCUSSION

**Training Dynamics.** We demonstrate the training curve and Pass@$k$ metric of Qwen-2.5-7B-Instruct trained by RFTT on challenging AIME-24 benchmark in Figure 5. As depicted, the entropy exhibits

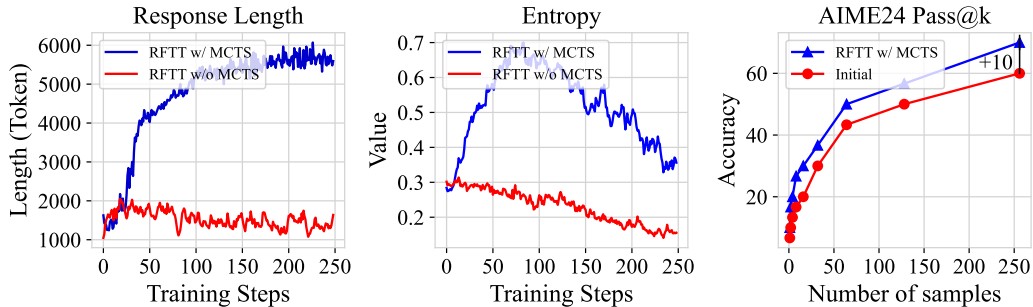

Figure 5: The training dynamics of RFTT on Qwen-2.5-7B-Instruct during RL.

a trend of first increasing then decreasing, suggesting a shift from diverse exploration at the early stage to the exploitation of high-quality reasoning paths in the later stage. The response length increases sharply within 50 training steps and then gradually increases when using PRM in RFTT. The searching time of the policy model undergoes continuous improvement during the RL training process, developing the capacity for exploration and optimization with appropriate incentives. We observe that as the search depth increases, the policy reinforces predefined human-like reasoning behavior, including the self-correction mechanism and the exploration of diverse problem-solving solutions, thereby evolving to tackle challenging tasks. Besides, the model trained after RFTT achieves a higher pass@$k$ score than the initial model when $k$ is large, which proves the effectiveness of our approach in promoting the model's exploration capability, achieving a higher capability ceiling finally. Please refer to Appendix G for more discussions.

**Computational Costs.** Detailed information regarding the computational resources for our method is provided in Table 7. The integration of Monte Carlo Tree Search (MCTS) into the Reinforcement Learning (RL) stage

Table 7: Comparison of computational cost.

| Method | Phase | Time (min) | Token Usage (k) |
|---|---|---|---|
| Data Construction | SFT | 13 min / question | 196 k / rollout |
| RFTT w/o MCTS | RL | 3.2 min / step | 511.4 k / step |
| RFTT (Ours) | RL | 12.1 min / step ↑× 3.8 | 216.8 k / step ↓ × 2.4 |

results in about four-fold increase in training time, despite a **reduction in the number of generated tokens** (due to shared prefixes) per training step. This overhead is attributed to the MCTS implementation, wherein the exploration of multiple search branches disrupts the batched and parallel nature of the standard sampling process. We anticipate that future optimizations to the MCTS sampling algorithm could mitigate this computational burden.

## 5 CONCLUSIONS

This work proposes RFTT, a reinforced fine-tuning framework that equips LLMs with advanced reasoning capabilities. By embedding learnable functional tokens into the model vocabulary, RFTT enables LLMs to internalize human-like reasoning behaviors, eliminating reliance on external prompts. Our approach advances the learning paradigm by bridging functional token-guided reasoning capability with diverse exploration during model training, offering a promising direction for developing resource-efficient, generalizable reasoning capabilities in smaller LLMs. Extensive experiments validate the effectiveness of RFTT, achieving significant performance gains and demonstrating scalability with inference-time search. Currently, our work is primarily training on mathematical reasoning tasks; its effectiveness on broader reasoning domains needs further exploration.

## ACKNOWLEDGEMENT

This work is supported in part by the Hangzhou Joint Funds of the Zhejiang Provincial Natural Science Foundation of China under Grant No. LHZSD24F020001, in part by the Zhejiang Province High-Level Talents Special Support Program "Leading Talent of Technological Innovation of Ten-Thousands Talents Program" under Grant No. 2022R52046, in part by the Fundamental Research Funds for the Central Universities under Grant No. 2021FZZX001-23, and in part by the advanced computing resources provided by the Supercomputing Center of Hangzhou City University. Dr Tao's research is partially supported by NTU RSR and Start Up Grants.

ETHICS STATEMENT

This paper presents work whose goal is to advance the field of machine learning, enhancing the reasoning capabilities of LLMs. There are many potential societal consequences of our work, none of which we feel will raise potential violations of the ICLR Code of Ethics.

REPRODUCIBILITY STATEMENT

To ensure reproducibility, we present the detailed experimental setup, including hardware, time, and hyperparameters of the MCTS procedures and two-phase training procedures in Section 4.2. The algorithm description in Section 3.2, Section 3.3, and Appendix D is provided in detail to ensure clarity and repeatability. The relevant code and dataset used in training are made public in this url.

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

# Appendix

TABLE OF CONTENTS

# A    MORE DISCUSSION OF RELATED WORKS

Drawing inspiration from AlphaGo (Silver et al., 2016), advanced self-play methods further integrate Monte Carlo Tree Search (MCTS) to explore high-quality reasoning paths by constructing reasoning trees (Hao et al., 2023; Zhang et al., 2024a;b; Guan et al., 2025). However, these direct tree-search approaches face two fundamental challenges: (1) LLMs tend to generate homogeneous reasoning paths due to their inherent preference for syntactic patterns established during training (Wei et al., 2022; Wang et al., 2023c; Patil, 2025); (2) the high-dimensional search space further limits the exploration capabilities of LLMs, as the combinatorial vocabulary space leads to an exponential growth in candidate reasoning paths (Zhang et al., 2024a).

A recent work rStar (Qi et al., 2024) introduces functional prompts (*e.g.*, "propose a one-step thought", "rephrase the question") that steer tree search by simulating human-like reasoning behaviors, thereby diversifying node exploration while constraining the search space. However, rStar operates purely as an inference-time augmentation without internalizing the reasoning capabilities through model training. The functional prompts act as external constraints rather than learned patterns. As a result, each inference of rStar must rely on iterative tree-search traversal across different functional prompts to seek correct solutions. Compared to direct reasoning methods (which directly generate CoT responses without relying on search), this exhaustive searching approach results in a substantial number of redundant LLM inference calls.

More recently, several studies (Hou et al., 2025; Shen et al., 2025) also incorporate searching mechanisms for RL training. TreeRL (Hou et al., 2025) first samples trajectories and then identifies tokens that have high entropies. After that, the model continues to generate the rest responses from the positions of these tokens. As we discussed before, LLMs tend to generate homogeneous reasoning paths due to their inherent preference for syntactic patterns established during pre-training. For advanced reasoning patterns such as self-reflection and self-correction that are not learned in pre-training, it is still difficult to explore them simply by sampling from high-entropy tokens. What's more, TreeRL indeed adopts vocabulary-level actions to conduct tree search on LLMs, which has a larger search space and reduced search efficiency. Satori (Shen et al., 2025) introduces three reasoning actions and adopts the Restart and Explore (RAE) algorithm. Specifically, RAE augments training data by randomly backtracking trajectories and inserting reflection tokens in their intermediate step. Its exploration is static and manually designed rather than adaptive to training dynamics. Moreover, random insertion of reflection tokens at intermediate steps may be suboptimal.

Therefore, we introduce functional tokens to form structured training data after MCTS, enabling the model to establish internalized token-guided reasoning patterns. After SFT, the model can perform diverse reasoning tree search by sampling the functional tokens in the model vocabulary without relying on any prompts during inference. This enables the model to automatically discover and reinforce optimal functional token combinations through reward signals, rather than relying on manual backtracking to restart exploration. Moreover, compared to other MCTS-based methods in Table 8, we additionally introduce cross verification and branch merging, enabling the connection of both correct and incorrect solution paths for more efficient reasoning paths.

Table 8: Performance of RFTT on wider reasoning domains.

| Method | different reasoning pattern | self-verification | self-correction |
|---|:---:|:---:|:---:|
| AlphaMath (Chen et al., 2024) | ✗ | ✗ | ✗ |
| AlphaLLM (Tian et al., 2024) | ✗ | ✗ | ✗ |
| ReST-MCTS* (Zhang et al., 2024a) | ✗ | ✗ | ✗ |
| TS-LLM (Feng et al., 2023) | ✗ | ✗ | ✗ |
| LE-MCTS (Park et al., 2024) | ✓ | ✗ | ✗ |
| rStar (Qi et al., 2024) | ✓ | ✗ | ✗ |
| MCTSr (Zhang et al., 2024b) | ✗ | ✗ | ✓ |
| **Ours** | ✓ | ✓ | ✓ |

Table 9: Accuracy of our proposed RFTT and baselines across different mathematical reasoning benchmarks. The best results in each box are highlighted in **bold**. The proposed RFTT significantly boosts the performance of smaller LLMs across all datasets.

| Model | Setting | In-Domain | Close-Domain | | | | AVERAGE |
|---|---|---|---|---|---|---|---|
| | | MATH-500 | GSM8K | SVAMP | Olympiad Bench | AMC | |
| LLaMA-3.2-3B-Instruct | Zero-shot CoT | 46.8 | 75.8 | 78.1 | 15.7 | 17.5 | 46.78 |
| | Few-shot CoT | 47.2 | 77.7 | 77.3 | 17.4 | 17.5 | 47.42 |
| | CoT+SC@4 | 49.8 | 79.5 | 80.5 | 19.6 | 22.5 | 50.38 |
| | ReFT | 52.4 | 78.5 | 81.6 | 22.8 | 35.0 | 54.06 |
| | **SFT Warmup (Ours)** | 50.6 | 77.4 | 79.1 | 20.6 | 35.0 | 52.54 |
| | **RFTT (Ours)** | **56.2** | **82.9** | **84.2** | **24.5** | **47.5** | **59.06** |
| Qwen-3-14B-Base | Zero-shot CoT | 68.4 | 89.6 | 80.7 | 37.9 | 62.5 | 67.82 |
| | Few-shot CoT | 75.8 | 92.5 | 82.1 | 40.5 | 67.5 | 71.70 |
| | CoT+SC@4 | 76.2 | 93.7 | 82.9 | 41.2 | 70.0 | 72.80 |
| | ReFT | 85.8 | 95.7 | 92.5 | 49.8 | 80.0 | 80.76 |
| | **SFT Warmup (Ours)** | 78.6 | 93.4 | 86.8 | 43.1 | 72.5 | 74.84 |
| | **RFTT (Ours)** | **89.0** | **97.2** | **95.3** | **53.0** | **85.0** | **83.90** |

Table 10: Comparison with entropy-based RL methods for promoting exploration. The best results in each box are highlighted in **bold**.

| Model | Setting | In-Domain | Close-Domain | | | | AVERAGE |
|---|---|---|---|---|---|---|---|
| | | MATH-500 | GSM8K | SVAMP | Olympiad Bench | AMC | |
| Qwen-3-4B-Base | Clip-Cov | 76.4 | 92.5 | 89.1 | 43.3 | 70.0 | 74.26 |
| | DAPO | 79.4 | 93.5 | 90.1 | 44.7 | 70.0 | 75.54 |
| | **RFTT** | 83.4 | 96.1 | 94.7 | 48.1 | 75.0 | 79.46 |
| | **RFTT + Clip-Cov** | 83.8 | 95.7 | 95.4 | 48.7 | 75.0 | 79.72 |
| | **RFTT + DAPO** | 85.0 | 96.4 | 95.2 | 49.8 | 77.5 | **80.78** |
| LLaMA-3.1-8B-Instruct | Clip-Cov | 54.6 | 89.3 | 84.4 | 25.0 | 27.5 | 56.16 |
| | DAPO | 56.4 | 90.1 | 84.5 | 25.3 | 27.5 | 56.76 |
| | **RFTT** | 60.2 | 91.9 | 87.5 | 29.8 | 55.0 | 64.88 |
| | **RFTT + Clip-Cov** | 60.8 | 92.6 | 88.2 | 30.4 | 55.0 | 65.40 |
| | **RFTT + DAPO** | 62.4 | 92.7 | 88.2 | 30.5 | 57.5 | **66.26** |

# B  ADDITIONAL RESULTS

**Different Model Sizes.** In Section 4.3, we provide the results on models with 4B, 7B, and 8B parameters. We further provide the performance of RFTT on models on LLaMA-3.2-3B-Instruct (Dubey et al., 2024) and Qwen-3-14B-Base (Yang et al., 2025) in Table 9 to validate the effectiveness of RFTT across different model backbones and parameter sizes.

**Iterative Improvement on Policy Models.** We conduct experiments on LLaMA-3.1-8B-Instruct for one iterative update on the MATH dataset. Specifically, we collect the explored reasoning paths with the correct answers in the first round RL phase, and further use them for the second round supervised fine-tuning on the policy $\pi_{\text{SFT}}$ once more. We introduce ReST-MCTS* as a comparable baseline, which uses iterative MCTS to collect training data for self-training. In Table 11, we list the results of iterative updates on two datasets, demonstrating a continuous enhancement of the initial policy's performance. Notably, our framework achieves superior results compared to ReST-MCTS* (7.9% vs. 1.7% improvement on MATH-500), despite the latter employing an extra reward model to guide its searching.

Table 11: Performance gains under self-improvement.

| Dataset | Method | 1st Iter | 2nd Iter | Ratio ↑ |
|---|---|---|---|---|
| GSM8K | ReST-MCTS* | 85.0 | 86.2 | 1.4% |
| | **Ours** | **86.4** | **88.1** | **2.0%** |
| MATH-500 | ReST-MCTS* | 47.6 | 48.4 | 1.7% |
| | **Ours** | **53.2** | **57.4** | **7.9%** |

**Comparison with entropy-based RL methods.** We further compare with DAPO (Yao et al., 2025) and Clip-Cov (Cui et al., 2025) that explicitly examined entropy mechanisms to enhance RL for exploration, as presented in Table 10. Note that DAPO adds a diversity term $J_{Div}(\pi_\theta)$ to the original RL objective to facilitate exploration, while Clip-Cov restricts gradient updates of tokens with excessively high covariance. RFTT is fundamentally orthogonal to these entropy-based RL tricks:

Table 12: Data information curated from non-mathematical reasoning domains.

| Data Source | MMLU-Pro (health) | MMLU-Pro (biology) | MMLU-Pro (physics) | MMLU-Pro (chemistry) | MMLU-Pro (law) | HumanEval |
|---|---|---|---|---|---|---|
| # Samples | 267 | 144 | 328 | 135 | 126 | 160 |

Table 13: Performance of Qwen-2.5-7B-Instruct being trained on general domain data.

| Settings | MMLU (STEM) | MMLU (Professional Medicine) | MMLU (Professional Law) | GPQA | CommonsenseQA | MBPP |
|---|---|---|---|---|---|---|
| Few-shot CoT | 81.1 | 77.6 | 50.4 | 36.4 | 81.6 | 79.2 |
| RFTT (Ours) | **86.9** | **83.1** | **55.8** | **38.8** | **85.2** | **83.8** |

they act at the RL optimization objective, while our method promotes exploration by improving the diversity of sampling. We operate at different levels but do not conflict with each other. The results show that adding these tricks to loss functions can further improve the performance of our method.

**Broader Generalization.** To further explore generalization, we have additionally conducted experiments using training data curated from non-mathematical general domains including MMLU-Pro (Wang et al., 2024c) and HumanEval (Chen et al., 2021a), as shown in Table 12. We evaluate the performance on three different reasoning domains: (1) *Commonsense* reasoning benchmarks including Medicine (MMLU (Hendrycks et al., 2021a)), Law (MMLU), and CommonsenseQA (Talmor et al., 2019); (2) *Science* reasoning benchmarks including STEM (MMLU) and GPQA (Rein et al., 2024); (3) *Code* generation benchmarks including MBPP (Austin et al., 2021). The results in Table 13 show broader improvements when RFTT is exposed to more diverse reasoning data, indicating strong potential for cross-domain generalization.

**Token Frequency.** We analyze the combinations of used functional tokens on MATH-500, GPQA, and MMLU. Table 14 shows that in math and science domain (MATH-500 and GPQA), the model tends to engage in analysis, reflection, and correction. While in commonsense reasoning (MMLU), it prefers to clarify the problem and then think step by step without reflection.

Table 14: Token usage in different reasoning domains.

| Dataset | Top-3 Combinations |
|---|---|
| MATH-500 | $a_2+a_3+a_4+a_6$, $a_2+a_3+a_4+a_5+a_6$, $a_2+a_3+a_4+a_6+a_7$ |
| GPQA | $a_2+a_3+a_4+a_5+a_6$, $a_1+a_3+a_4$, $a_1+a_2+a_4+a_6+a_7$ |
| MMLU | $a_1+a_3+a_4$, $a_1+a_3+a_4+a_5$, $a_1+a_3+a_4+a_6$ |

**Reward hacking analysis.** In practice, we combine outcome-based reward supplemented by PRM score signals, which aims to balance between both accuracy and quality of the reasoning process. We further analyzed the average process rewards (ranging from -0.5 to 0.5) of trajectories generated by models on the MATH-500 dataset. The results in Table 15 show that the average PRM scores of incorrect trajectories are much lower than those of correct trajectories, which may alleviate the problem of reward hacking.

Table 15: Potential reward hacking analysis when rewarding reasoning chains.

| Avg. Score | Correct Samples | Incorrect Samples |
|---|---|---|
| Qwen-3-4B-Base | 0.38 | -0.04 |
| Qwen-2.5-7B-Instruct | 0.34 | 0.02 |
| LLaMA-3.1-8B-Instruct | 0.31 | -0.05 |

## C EXPERIMENTAL DETAILS

**Dataset Information.** We list detailed information of our training and evaluation dataset in Table 17. Specifically, we select 1000 and 3994 questions from the training set of MATH for SFT and RL, respectively. Recent studies have revealed that focusing on more difficult problems can better enhance reasoning capability (Min et al., 2024b; Team et al., 2025), thus we increase the proportion of difficult questions (Level 4 and Level 5) in our training. The data distribution of SFT and RL phases are shown in Figure 6.

**Evaluation Method**. In evaluating the performance of superior LLMs, we take the relevant metrics from their official technical reports (Achiam et al., 2023; Anthropic, 2024; Liu et al., 2024a; Dubey et al., 2024; Yang et al., 2024a). For non-reported metrics, we report their accuracies on corresponding

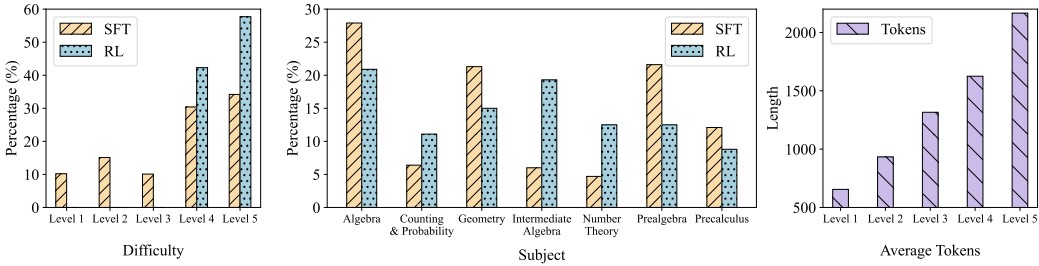

Figure 6: Detailed statistical information of the training dataset in two-phase training.

Table 16: The CoT template for evaluation. The variable question will be replaced with the specific problem when testing.

---

You are a mathematics expert that solves problems step by step.
Note that you should output your final answer with the format of "The answer is: boxed{<ANSWER>}.",
where <ANSWER> should be a numeric result or a math expression.
Question: {question}
Let's think step by step.

---

datasets using the official API with zero-shot (original question without any prompts). For the results of CoT, we demonstrate the CoT prompt template used to evaluate in Table 16.

## D  MCTS PROCEDURE

The MCTS algorithm is capable of balancing exploration and exploitation within vast environmental spaces, enabling it to quickly identify high-quality steps or trajectories. Due to its focus on exploring unseen states and sampling high-quality trajectories, which is a key aspect of online sampling in reinforcement learning, MCTS has been widely applied in reinforcement learning tasks, such as AlphaGo and AlphaZero (Silver et al., 2018). The standard MCTS algorithm consists of four stages: *selection*, *expansion*, *simulation*, and *backpropagation*.

(1) **Selection.** Starting from root node (question) $s_0 = x$, a series of internal nodes $\{s_1, s_2, ..., s_l\}$ are chosen until reaching a leaf node $s_l$. In the selection phase, we use the Upper Confidence Bound applied to trees (UCT) to select each node for balancing exploration and exploitation:

$$a_t = \begin{cases} \arg\max_{a \in U(s_t)} \pi_\theta(a|s_{0:t}), & U(s_t) \neq \emptyset, \\ \arg\max_{a \in \mathcal{A}(s_t)} \texttt{UCT}(s_t, a), & \text{otherwise,} \end{cases} \tag{5}$$

$$\texttt{UCT}(s_t, a) = \frac{Q(s_t, a)}{N(s_t, a)} + c \cdot \sqrt{\frac{\ln N(s_t)}{N(s_t, a)}}, \tag{6}$$

where $\mathcal{A}(s_t)$ represents the set of possible actions (functional tokens) that can be selected after the current step $s_t$, $U(s_t)$ denotes unexplored functional tokens after the current step $s_t$, and $\pi_\theta(a|s_{0:t})$ is the probability to sample functional token $a$ based on the current state $s_{0:t}$. In UCT score, $N(s_t)$ denotes the number of times step $s_t$ has been visited in previous iterations, $N(s_t, a)$ denotes the number of times action $a$ has been selected after step $s_t$, $Q(s_t, a)$ is the cumulative reward value of selecting action $a$ after step $s_t$, and $c$ is a constant that balances exploitation and exploration.

In LLM sampling, each action in MCTS corresponds to a reasoning step. The sequence of generated steps forms the current state. Selecting an action in MCTS means generating the next step, and the next state is obtained by appending this step to the already generated steps.

(2) **Expansion.** If the selected leaf node $s_l$ is not terminal node, it is expanded by sampling an action from $\mathcal{A} = \{a_1, a_2, a_3, a_4, a_5, a_8\}$. We then use the action $a_i \in \mathcal{A}$ (functional prompt in SFT or functional token in RL) to guide the LLM to generate a corresponding reasoning step. Notably, the expansion of certain actions has dependency relationships for efficient exploration. For example, <clarify> ($a_1$) can only be expanded after the root question, and <subquestion> ($a_3$) can only expand the <next_step> ($a_4$) and <direct_answer> ($a_5$). All prerequisite dependencies are presented in Table 18.

Table 17: The detailed information of five datasets.

| Dataset | # Train | # Test | # Domain |
|---------|---------|--------|----------|
| MATH | 7500 | 5000 | In-Domain |
| GSM8K | 7473 | 1319 | Out-of-Domain |
| SVAMP | - | 1000 | Out-of-Domain |
| Olympiad Bench | - | 675 | Out-of-Domain |
| AMC | - | 40 | Out-of-Domain |
| GPQA | - | 448 | Out-of-Domain |
| MMLU-Pro | - | 12032 | Out-of-Domain |

Table 18: The dependencies between actions.

| Previous Action | Next Action |
|-----------------|-------------|
| clarify ($a_1$) | $a_2, a_3, a_4, a_5$ |
| analysis ($a_2$) | $a_3, a_4, a_5$ |
| subquestion ($a_3$) | $a_4, a_5$ |
| next_step ($a_4$) | $a_3, a_4, a_5, a_6, a_8$ |
| direct_answer ($a_5$) | $a_6, a_8$ |
| verify ($a_6$) | $a_3, a_4, a_5, a_6, a_7, a_8$ |
| refine ($a_7$) | $a_3, a_4, a_5, a_6, a_8$ |

(3) **Simulation.** The simulation phase involves randomly selecting an action from expanded node, generating the corresponding step, and continuing this process until a terminal node is reached. We define the terminal node as achieving either the final answer or the maximum permitted depth.

(4) **Backpropagation.** Based on the results of the simulation, backpropagation updates the $Q$ and $N$ values for all nodes along the path from the expanded node to the root node as follows:

$$Q(s_t, a) \leftarrow Q(s_t, a) + r, \tag{7}$$

$$N(s_t, a) \leftarrow N(s_t, a) + 1, \quad N(s_t) = \sum_{a \in \mathcal{A}(s_t)} N(s_t, a) \tag{8}$$

where $r$ is the terminal reward obtained from the simulation reaching the terminal node.

## E  FUNCTIONAL TOKENS WITH PROMPTS

We present the functional tokens and prompts that were designed for our experiments.

- **`<clarify>`** ($a_1$): Organize the conditions of the given complex question and express them in a clearer form to avoid misunderstanding.
- **`<analysis>`** ($a_2$): Analyze the general problem-solving idea and the relevant knowledge.
- **`<subquestion>`** ($a_3$): Break down complex reasoning problems into manageable sub-problems.
- **`<next_step>`** ($a_4$): Propose the next intermediate step based on the existing reasoning steps.
- **`<direct_answer>`** ($a_5$): Complete the remaining steps step-by-step to reach the final answer.
- **`<verify>`** ($a_6$): Reflect if there are any mistakes in the preceding reasoning steps, considering LLMs might make computational or logical errors in a reasoning step.
- **`<refine>`** ($a_7$): Correct certain previous steps if the errors are identified, ensuring that each step is accurate to achieve a final outcome.
- **`<output>`** ($a_8$): Output the final answer that is in the expected format.

---

**Prompt for Functional Token `<clarify>`**

You are an AI assistant to help me clarify questions by restating the original question and task in a clear and comprehensive manner. In your clarified version, ensure that all information from the original question is fully expressed. Following are some useful examples.

Original Question: Increasing the radius of a cylinder by 6 units increased the volume by $y$ cubic units. Increasing the height of the cylinder by 6 units also increases the volume by $y$ cubic units. If the original height is 2, then what is the original radius?

Clarified Question: We are given a problem involving a cylinder where increasing the radius by 6 units and increasing the height by 6 units each results in an increase in volume by $y$ cubic units. Our goal is to find the original radius of the cylinder, given that the original height is 2.

Original Question: A wooden model of a square pyramid has a base edge of 12 cm and an altitude of 8 cm. A cut is made parallel to the base of the pyramid that separates it into two pieces: a smaller pyramid and a frustum. Each base edge of the smaller pyramid is 6 cm and its altitude is 4 cm. How many cubic centimeters are in the volume of the frustum?

---

Clarified Question: Let's understand what is being asked. Given a wooden model of a square pyramid has a base edge of 12 cm and an altitude of 8 cm. Also given a cut parallel to the base creates a smaller pyramid and a frustum. Note that the smaller pyramid has a base edge of 6 cm and an altitude of 4 cm. We need to find the volume of the frustum.

Question: { uesr_question }

### Prompt for Functional Token `<analysis>`

Given a math problem and an existing incomplete solution, your task is to provide a brief insight for the following problem. Note that it's not required to provide any specific solution process or calculation, but give a general guidance of sovling this problem. Usually one or two sentences are enough.

Question: { uesr_question }

### Prompt for Functional Token `<subquestion>`

Given a math problem and an existing incomplete solution, please propose a sub-problem that can be solved independently based on my current progress. The output format is limited to: "Let's ... now". Where ... indicates the omitted sub-problem that you should fill in. Note that do not propose sub-problems that have already been solved in previous steps.

Here is the input, please follow the restricted output format.

Question: { uesr_question }

### Prompt for Functional Token `<next_step>`

Given a math problem and an existing incomplete solution, your task is to propose one next step in a smooth and proper way.

If no existing steps are provided, you need to briefly analyse the problem from scratch and then output the first step. Otherwise, you need to output the correct next step of the existing solution, following the ideas of the existing steps.

Your output should be a single reasoning step that may include reasoning, detailed calculations process, choosing answers, etc.

Question: { uesr_question }

### Prompt for Functional Token `<direct_answer>`

Given a math problem and an existing incomplete solution, your task is to complete the remaining solution in a smooth and proper way. You need to give step-by-step solutions to the problem following the ideas of the existing steps, and do not repeat any existing steps.

Question: { uesr_question }

### Prompt for Functional Token `<verify>`

Given a problem and its previous solution steps, there is a correct subsequent solution and your subsequent solution. Please meticulously review your subsequent solution step by step and identify if any mistakes occur.
# Correct solution
{correct_solution}
# Your subsequent solution
{solution}

# F    REASONING CASES

Our constructed data consists of diverse human-like reasoning behaviors, including double check, self-reflection, and self-correction. The following are some data examples of diverse reasoning paths in mathematical reasoning.

---

**Case 1: Self-Verification**

**# Problem:** Complex numbers $a$, $b$, $c$ form an equilateral triangle with side length 18 in the complex plane. If $|a + b + c| = 36$, find $|ab + ac + bc|$.

**# Output:** $\langle$clarify$\rangle$ The complex numbers $a$, $b$, and $c$ form an equilateral triangle with side length 18 in the complex plane. Additionally, we know that $|a+b+c| = 36$. Our task is to find the value of $|ab+ac+bc|$. $\langle$/clarify$\rangle$

$\langle$analysis$\rangle$ Use the properties of equilateral triangles in the complex plane, where the centroid (average) of the vertices is involved, and relate it to the given magnitude. Utilize the fact that for an equilateral triangle formed by complex numbers $a$, $b$, and $c$, the expression $a^2 + b^2 + c^2 = ab + ac + bc$. $\langle$/analysis$\rangle$

$\langle$subquestion$\rangle$ First, Let me find the value of $a + b + c$. $\langle$/subquestion$\rangle$

$\langle$next_step$\rangle$ Since the centroid of the equilateral triangle formed by $a$, $b$, and $c$ is given by $\frac{a+b+c}{3}$, and we know $|a + b + c| = 36$, we can find $|a + b + c|$ directly as 36. However, the next logical step is to use this information to find $a + b + c$.

Given $|a + b + c| = 36$, we can denote $a + b + c = z$, where $|z| = 36$. The specific process for this step is to recognize that $a + b + c$ is a complex number with magnitude 36, but without loss of generality, we can assume $a + b + c = 36e^{i\theta}$ for some angle $\theta$. For simplicity, we can consider the case where $\theta = 0$, so $a + b + c = 36$. This assumption simplifies the calculation without affecting the final result due to the properties of magnitudes.

So, we have:
$$a + b + c = 36.$$

$\langle$/next_step$\rangle$

$\langle$subquestion$\rangle$ Next, I need to find the value of $|ab + ac + bc|$. $\langle$/subquestion$\rangle$

$\langle$next_step$\rangle$ Moreover, I will use the identity for the sum of products of roots of a polynomial.

$$|ab + ac + bc| = \left| \frac{(a+b+c)^2 - (a^2 + b^2 + c^2)}{2} \right|.$$

$\langle$/next_step$\rangle$

$\langle$subquestion$\rangle$ Then, we must calculate $a^2 + b^2 + c^2$. $\langle$/subquestion$\rangle$

$\langle$next_step$\rangle$ Using the property of equilateral triangles in the complex plane, we know that $a^2 + b^2 + c^2 = ab + ac + bc$. However, we can also express $a^2 + b^2 + c^2$ in terms of $a + b + c$ and $ab + ac + bc$:

$$a^2 + b^2 + c^2 = (a+b+c)^2 - 2(ab + ac + bc).$$

Given $a + b + c = 36$, we substitute to get:

$$a^2 + b^2 + c^2 = 36^2 - 2(ab + ac + bc).$$

Now, let's solve for $ab + ac + bc$. $\langle$/next_step$\rangle$

$\langle$subquestion$\rangle$ To continue, I need to solve for $ab + ac + bc$. $\langle$/subquestion$\rangle$

$\langle$next_step$\rangle$ We already have:

$$a^2 + b^2 + c^2 = 36^2 - 2(ab + ac + bc).$$

And since $a^2 + b^2 + c^2 = ab + ac + bc$, we substitute to get:

$$ab + ac + bc = 36^2 - 2(ab + ac + bc).$$

Rearrange to solve for $ab + ac + bc$:

$$3(ab + ac + bc) = 36^2,$$

$$ab + ac + bc = \frac{36^2}{3} = \frac{1296}{3} = 432.$$

Now, let's find $|ab + ac + bc|$. $\langle$/next_step$\rangle$

$\langle$verify$\rangle$ Hold on, let me confirm that. The given steps correctly use the properties of equilateral triangles in the complex plane, including that the centroid (average) of the vertices is involved and the identity $a^2 + b^2 + c^2 = ab + ac + bc$. The calculation for $ab + ac + bc$ is correct:

$$3(ab + ac + bc) = 36^2,$$

---

$$ab + ac + bc = \frac{1296}{3} = 432.$$

The magnitude of a real number is the number itself, so $|ab + ac + bc| = 432$. There are no mistakes in the logic or calculations. $\langle$/verify$\rangle$

$\langle$output$\rangle$ So the answer is $\boxed{432}$. $\langle$/output$\rangle$

---

### Case 2: Self-Reflection and Self-Correction

# Problem: Suppose that $f(x) = \frac{1}{2x+b}$. For what value of $b$ does $f^{-1}(x) = \frac{1-2x}{2x}$?"

# Output: $\langle$analysis$\rangle$ To find the value of $b$ for which the inverse function $f^{-1}(x) = \frac{1-2x}{2x}$ matches with $f(x) = \frac{1}{2x+b}$, we start by understanding that if $y = f(x)$, then $x = f^{-1}(y)$. This means substituting $f(x)$ into the given form of $f^{-1}(x)$ should yield $x$. $\langle$/analysis$\rangle$

$\langle$next_step$\rangle$ First, substitute $f(x) = \frac{1}{2x+b}$ into the equation for $f^{-1}(x)$, giving us:

$$x = \frac{1 - 2\left(\frac{1}{2x+b}\right)}{2\left(\frac{1}{2x+b}\right)}.$$

This is based on the principle that applying a function and its inverse in succession returns the original input, i.e., $f^{-1}(f(x)) = x$. $\langle$/next_step$\rangle$

$\langle$next_step$\rangle$ Simplify the right-hand side:

$$x = \frac{1 - \frac{2}{2x+b}}{\frac{2}{2x+b}} = \frac{(2x + b) - 2}{2} = \frac{2x + b - 2}{2}.$$

The simplification involves combining terms and recognizing that dividing by a fraction is equivalent to multiplying by its reciprocal. $\langle$/next_step$\rangle$

$\langle$next_step$\rangle$ For the expression to equal $x$, we set the simplified form equal to $x$:

$$x = \frac{2x + b - 2}{2}.$$

To solve for $b$, we equate the numerators (since denominators are already the same and non-zero), leading to:

$$2x = 2x + b - 2.$$

$\langle$/next_step$\rangle$

$\langle$direct_answer$\rangle$ From the equation $2x = 2x + b - 2$, subtract $2x$ from both sides to isolate $b$:

$$0 = b - 2,$$

which simplifies to:

$$b = 2.$$

Thus, the value of $b$ that satisfies the condition is $b = 2$, confirming the expected answer. $\langle$/direct_answer$\rangle$

$\langle$verify$\rangle$ But I have a feeling that it's a little complex. Maybe I should consider a simpler approach. $\langle$/verify$\rangle$

$\langle$analysis$\rangle$ To find the value of $b$, first express $f^{-1}(x)$ in terms of $f(x)$ and then solve for $b$. $\langle$/analysis$\rangle$

$\langle$subquestion$\rangle$ First, I should solve for $f^{-1}(x)$ in terms of $x$. $\langle$/subquestion$\rangle$

$\langle$next_step$\rangle$ Start with $y = f(x) = \frac{1}{2x+b}$. Swap $x$ and $y$ to get $x = \frac{1}{2y+b}$. Then solve for $y$ to find $f^{-1}(x)$. $\langle$/next_step$\rangle$

$\langle$subquestion$\rangle$ Next, I need to solve for $y$ in the equation $x = \frac{1}{2y+b}$. $\langle$/subquestion$\rangle$

$\langle$next_step$\rangle$ Multiply both sides by $2y + b$ to get rid of the denominator, then isolate $y$. $\langle$/next_step$\rangle$

$\langle$next_step$\rangle$ Multiply both sides by $2y + b$ to obtain $x(2y + b) = 1$. Then distribute $x$ on the left side to get $2xy + bx = 1$. $\langle$/next_step$\rangle$

$\langle$direct_answer$\rangle$ Next, we will isolate $y$.

Starting from the equation $2xy + bx = 1$:

$$2xy + bx = 1$$

Subtract $2xy$ from both sides:

$$bx = 1 - 2xy$$

Divide both sides by $b$:

$$y = \frac{1 - 2xy}{b}$$

Since we know that $f^{-1}(x) = \frac{1-2x}{2x}$, we can equate the expressions for $y$:

$$\frac{1 - 2xy}{b} = \frac{1 - 2x}{2x}$$

For the above equality to hold for all $x$, the numerators and denominators must match. Therefore, we can compare the denominators directly:

$$b = 2x \cdot 2x = 2 \cdot 2 = 4$$

$\langle$/direct_answer$\rangle$
$\langle$verify$\rangle$ Oops, I seem to get two different results. Actually, I'm getting confused. I think I should review carefully about my previous solutions.
The mistake occurs when dividing by $b$ and incorrectly expressing $y$ as $y = \frac{1-2xy}{b}$.
The correct form should be $y = \frac{1-bx}{2x}$.
As a result, the value of $b$ is incorrectly determined. $\langle$/verify$\rangle$
$\langle$refine$\rangle$ Correct the expression for $y$ after isolating it: $y = \frac{1-bx}{2x}$.
Equate this to the given $f^{-1}(x) = \frac{1-2x}{2x}$:

$$\frac{1 - bx}{2x} = \frac{1 - 2x}{2x}$$

Since the denominators are the same, equate the numerators:

$$1 - bx = 1 - 2x$$

- Subtract 1 from both sides:

$$-bx = -2x$$

- Divide both sides by $-x$ (assuming $x \neq 0$):

$$b = 2$$

Thus, the value of $b$ is 2. $\langle$/refine$\rangle$
$\langle$output$\rangle$ The answer is: $\boxed{2}$. $\langle$/output$\rangle$

We provide the following response of Qwen-2.5-7B-Instruct comparisons before and after training to demonstrate the effectiveness of our method.

### Problem from MATH

Semicircles of diameter 2" are lined up as shown. What is the area, in square inches, of the shaded region in a 1-foot length of this pattern? Express your answer in terms of $\pi$.

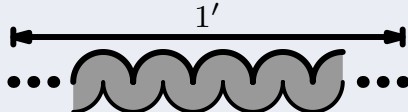

### Initial Response of Qwen-2.5-7B-Instruct

To determine the area of the shaded region in a 1-foot length of the given pattern, we need to analyze the pattern and calculate the area of the semicircles involved.
First, note that 1 foot is equal to 12 inches. The pattern consists of semicircles of diameter 2 inches, which means each semicircle has a radius of 1 inch.
The area of one semicircle is given by: Area of one semicircle $= \frac{1}{2}\pi r^2 = \frac{1}{2}\pi(1)^2 = \frac{1}{2}\pi$.
Next, we need to determine how many semicircles fit into a 12-inch length. Since each semicircle has a diameter of 2 inches, the number of semicircles in 12 inches is $\frac{12}{2} = 6$.

The pattern alternates between semicircles above and below the horizontal line, forming a repeating sequence of 6 semicircles. The shaded region consists of the top 3 semicircles and the bottom 3 semicircles. The total area of the shaded region is the sum of the areas of these 6 semicircles: Total area of shaded region $= 6 \times \frac{1}{2}\pi = 3\pi$.

Thus, the area of the shaded region in a 1-foot length of the pattern is $\boxed{3\pi}$.

**Response Trained by RFTT**

$\langle$clarify$\rangle$ We are given a pattern where semicircles of diameter 2 inches are lined up in a specific arrangement. The task is to find the area, in square inches, of the shaded region within a 1-foot (which is 12 inches) length of this pattern. The answer should be expressed in terms of $\pi$. $\langle$/clarify$\rangle$

$\langle$next_step$\rangle$ First, I need to calculate the radius of each semicircle. The diameter of each semicircle is 2 inches, so the radius is half of the diameter, which is 1 inch. $\langle$/next_step$\rangle$

$\langle$subquestion$\rangle$ Next, Let me calculate the area of one semicircle. $\langle$/subquestion$\rangle$

$\langle$next_step$\rangle$ The area of a full circle with radius $r = 1$ inch is $\pi r^2 = \pi(1)^2 = \pi$ square inches. Therefore, the area of one semicircle is half of that, which is $\frac{\pi}{2}$ square inches. $\langle$/next_step$\rangle$

$\langle$subquestion$\rangle$/ Moreover, I could attempt to determine how many semicircles fit into a 12-inch length. $\langle$/subquestion$\rangle$

$\langle$next_step$\rangle$ The pattern consists of alternating upper and lower semicircles, each with a diameter of 2 inches (radius 1 inch). Upper semicircles curve upward, and lower semicircles curve downward, forming a repeating wavy pattern. The pattern repeats every 2 inches. In each 2-inch segment, there is one upper semicircle and one lower semicircle. $\langle$/next_step$\rangle$

$\langle$next_step$\rangle$ Then, I could attempt to calculate the total area of semicircles in 12 inches. In each 2-inch segment, the shaded area is the sum of the area of one upper semicircle and one lower semicircle: ($\frac{\pi}{2} + \frac{\pi}{2} = \pi$). Since the pattern repeats every 2 inches, a 12-inch length contains ( $\frac{12}{2} = 6$ ) such segments. Thus, the total area of the shaded region is ( $6 \times \pi = 6\pi$ ) $\langle$/next_step$\rangle$

$\langle$verify$\rangle$ Wait, the question arises: have we correctly calculated what comprises the "shaded" region? Upon reassessment, we acknowledge each row aligns exactly with the previous, consisting of precisely half periods in overlapped symmetry, with no shaded region lawyer above certain baseline symmetry points. $\langle$/verify$\rangle$

$\langle$output$\rangle$ So the answer is $\boxed{6\pi}$. $\langle$/output$\rangle$

## G FURTHER DISCUSSION AND LIMITATIONS

### G.1 HOW DO FUNCTIONAL TOKENS BENEFIT?

There are two main challenges in applying MCTS during training (Guo et al., 2025): (1) Unlike traditional games with well-defined action spaces, the combinatorial nature of vocabulary leads to exponential growth in candidate reasoning paths; (2) Training a fine-grained value model to effectively guide the search process remains highly challenging. The introduction of functional tokens with rich semantics can diversify node exploration while constraining the search space. Notably, we can use the intrinsic probability of functional tokens, which adjust dynamically during training to guide its search process, thereby estimating the need for training an extra value model. Furthermore, the functional tokens naturally divide the reasoning process into a series of fine-grained steps, which can seamlessly integrate with PRM.

### G.2 IS SFT WARMUP WITH FUNCTIONAL TOKENS NEEDED?

Recently, Deepseek R1-Zero (Guo et al., 2025) has demonstrated the emergence of self-reflection in the LLM's response during pure RL phase. We also encounter the occurrence of this kind of feature when conducting pure RL on Qwen-2.5-7B-Instruct (RFTT w/o SFT Warmup), as shown in Table 19. But such a pattern occurs at an extremely low rate (about 1.5%) on Qwen-2.5-7B-Instruct and there is even no such case observed on LLaMA-3.1-8B-Instruct. The power of RL is to reinforce high-quality reasoning patterns like self-reflection and exploration of different approaches, thus maximizing expected rewards by leveraging increased test-time computation. However, in a model with smaller parameters, the limitation arises from their inherent preference for syntactic patterns established in pretraining, which further restricts the reinforcement of sophisticated reasoning patterns. Therefore, SFT warmup with functional tokens play an important role in faster convergence and stable training.

Table 19: Emergence of self-reflection and self-correction during pure RL on Qwen-2.5-7B-Instruct.

| |
|---|
| Question: In $\triangle RST$, shown, $\sin R = \frac{2}{5}$. What is $\sin T$? [asy]...[/asy] |
| Response: Given that $\triangle RST$ is a right triangle with the right angle at $S$, we know that $\sin R = \frac{\text{opposite}}{\text{hypotenuse}} = \frac{RT}{RS}$. Here, $RT = \sqrt{21}$ and $RS = 5$. Therefore, we have:$\sin R = \frac{\sqrt{21}}{5} = \frac{2}{5}$. This implies that $\sqrt{21} = 2$, which is incorrect based on the given information. Instead, we should use the correct value of $\sin R = \frac{2}{5}$ and find the correct lengths. Let's denote the length of $RT$ as $a$ and the length of $RS$ as $c = 5$. Then, we have: $\sin R = \frac{a}{c} = \frac{a}{5} = \frac{2}{5} \implies a = 2$. Now, we can find the length of $ST$ using the Pythagorean theorem:$ST = \sqrt{RS^2 - RT^2} = \sqrt{5^2 - 2^2} = \sqrt{25 - 4} = \sqrt{21}$. In $\triangle RST$, $\sin T = \frac{\text{opposite}}{\text{hypotenuse}} = \frac{RS}{RT} = \frac{5}{2}$. However, we need to re-evaluate the correct trigonometric relationship. Since $\sin R = \frac{2}{5}$, we know that $\cos R = \sqrt{1 - \sin^2 R} = \sqrt{1 - \left(\frac{2}{5}\right)^2} = \sqrt{1 - \frac{4}{25}} = \sqrt{\frac{21}{25}} = \frac{\sqrt{21}}{5}$. In a right triangle, $\sin T = \cos R$. Therefore: $\sin T = \cos R = \frac{\sqrt{21}}{5}$. 

 Thus, the value of $\sin T$ is: $\boxed{\dfrac{\sqrt{21}}{5}}$. |

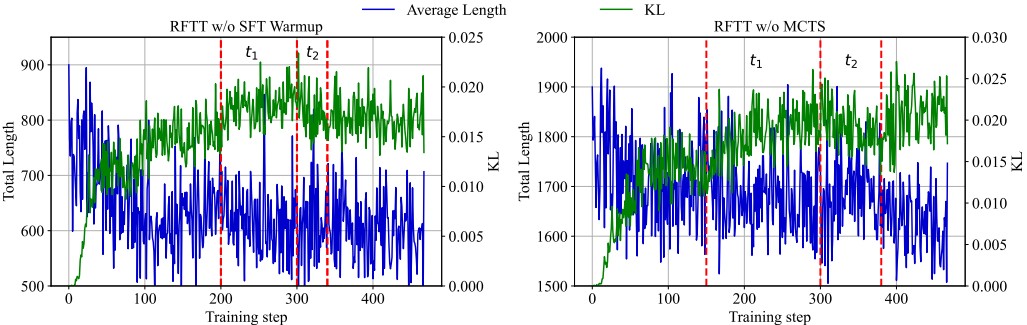

Figure 7: The training curve of RFTT w/o SFT Warmup and w/o MCTS during RL.

## G.3 DISCUSSION OF RESPONSE LENGTHS.

Recent research suggests that the reasoning capabilities of the model may benefit from longer response lengths (Guo et al., 2025). In Figure 7, we present the training curve of RFTT w/o SFT Warmup and RFTT w/o MCTS on LLaMA-3.2-3B-Instruct as an example. We conduct RFTT w/o SFT Warmup with the prompt template in Table 16. Surprisingly, we find that after an initial drop in the response lengths of the policy model, the lengths rebound but persistently remain below the response lengths observed in the original pretrained model. Moreover, we observe that as the response lengths drop, the KL divergence between the current policy and the initial model increases during training period $t_1$; while the lengths increase, the KL divergence drops during training period $t_2$. This indicates a potential correlation between response lengths and KL constraints. It seems that the subsequent increase in response lengths cannot be attributed to the exploration of sophisticated behaviors, but rather primarily stems from the impact of the KL penalty.

In contrast, the response lengths of RFTT w/o PRM in Figure 5 do not exhibit a downward trend. This is likely due to the benefits of our functional token-guided tree search, which provides a diverse exploration space. Furthermore, the use of PRM in MCTS introduces fine-grained reward assignments in intermediate functional tokens. Such a mechanism significantly mitigates the exploration limitations imposed by the KL divergence penalty, potentially elucidating the observed length expansion in RFTT with PRM during the training process. This highlights the effectiveness of RFTT, where the functional tokens serve as a bridge that seamlessly combines the employment of MCTS and PRM for learning to reason.

## G.4 LIMITATIONS

While our method improves reasoning via functional tokens and reinforcement learning, it may also introduce new risks. For example, the use of reasoning tokens could be exploited through token injection attacks to manipulate model behavior. Additionally, self-improvement via reinforcement learning raises concerns about unintended capability amplification or instability. More generally, enhanced reasoning ability does not eliminate common LLM issues such as bias, overconfidence, or the generation of incorrect but convincing outputs. We acknowledge these risks and believe they warrant further study as functional reasoning methods evolve.

