# OpenReview forum: "Incentivizing LLM Reasoning via Reinforcement Learning with Functional Monte Carlo Tree Search"
_ICLR.cc/2026/Conference — ICLR 2026 Poster_

### Official Review · Reviewer_xic9 · 2025-10-29

**Soundness:** 3
**Presentation:** 3
**Contribution:** 3
**Rating:** 6
**Confidence:** 3

**Summary:**

The paper proposes Reinforced Functional Token Tuning (RFTT), a novel reinforcement-based fine-tuning framework that equips large language models with learn-to-reason capabilities. Unlike prior prompt-driven reasoning methods, RFTT introduces learnable functional tokens (e.g., <analyze>, <verify>, <refine>) directly into the model’s vocabulary to enable structured and human-like reasoning behaviors. The framework consists of two stages: supervised fine-tuning with functional tree search to teach token usage, and online reinforcement learning to encourage autonomous exploration of reasoning pathways. Experimental results on mathematical and general reasoning benchmarks demonstrate that RFTT significantly outperforms baselines and continues to improve with increased test-time computation.

**Strengths:**

1. The introduction of learnable functional tokens provides an interpretable and flexible mechanism for modeling reasoning behaviors within the LLM’s vocabulary.
2. The two-phase framework combining supervised functional tree search with reinforcement learning.
3. Experiments demonstrate consistent and significant improvements on mathematical and general reasoning benchmarks

**Weaknesses:**

1. The training framework may computationally expensive due to the integration of Monte Carlo Tree Search, which limits scalability to larger models.
2. The semantics of functional tokens are predefined by design, which may constrain flexibility and introduce human bias into the reasoning process.
3. Rewarding reasoning chains may cause reward hacking, where the model learns to produce text that merely appears logically sound rather than genuinely correct.

**Questions:**

1. Please provide a analysis of the potential **reward hacking** problem that may arise when rewarding reasoning chains, where the LLM could exploit the reward model by generating answers that align with its biases or merely appear correct to maximize the reward.
2. How sensitive is RFTT to the choice and number of functional tokens? Would the performance drop if some tokens (e.g., <verify>, <refine>) were removed or replaced?
3. During functional tree search, what criteria are used to select the optimal reasoning branch?
4. How to get the dependencies between actions?

---

> ### Author Response · Authors · 2025-11-22
> **Response (1): W1 to Q2**
>
> We greatly appreciate the reviewer for your insightful comments and positive support.
>
> **[W1] Computational costs and scalability to larger models.**
>
> Since current LLM inference engines do not offer special optimizations for tree search, meaning the proposed RFTT still requires iterative search, resulting in a performance that is approximately 4× slower than random sampling (Table 7). However, our method reduces token budget by 2.4× benefits from shared prefixes. To further validate scalability, we present results on Qwen3-14B-Base in Table R1. The results show that our method achieves competitive performance when applied to larger models. *These additional results have been updated in Appendix B (Page 19) of our revised manuscript.*
>
> Table R1.
> |Models|Settings|MATH-500|GSM8K|SVAMP|Olympiad Bench|AMC-23|Avg.|
> |---|---|---|---|---|---|---|---|
> |Qwen3-14B-Base|Few-shot CoT|75.8|92.5| 82.1| 40.5| 67.5| 71.70|
> ||ReFT|85.8|95.7| 92.5| 49.8| 80.0| 80.76|
> ||RFTT(Ours)|89.0| 97.2| 95.3| 53.0| 85.0| **83.90**|
>
> **[W2 & Q2] The design and choice of functional tokens.**
>
> (1) Although our functional tokens are predefined, they are not only learned during SFT Warmup, but also further optimized during online RL. In the RL phase, the policy model dynamically selects actions (functional tokens) to generate corresponding reasoning steps, learns which action combinations lead to golden answers based on reward signals, and adaptively adjusts action probabilities. Therefore, the model can learn to identify optimal actions in different states on its own.
>
> (2) To validate the performance impact of each functional token, we mask the corresponding functional token (by not sampling it at test time) after learning all functional tokens into the model (Qwen2.5-7B-Instruct) at training. For your convenience, we include again here the results from Table 5 in our manuscript as Table R2. These results in Table R2 of our manuscript demonstrates the complementary effectiveness of the designed tokens. Notably, tokens like \<verify\> and \<refine\> have a significant contribution, and their absence leads to performance degradation. Furthermore, we remove the trajectories with tokens \<verify\>, \<refine\> and conduct RFTT two phase training. The results in Table R3 shows slight performance degradation without adding them in training.
>
> Table R2.
> |Masked Functional Token|Accuracy (%)|
> |-|-|
> |w/o \<clarify\>|79.0|
> |w/o \<analysis\>|77.6|
> |w/o \<subquestion\>|79.4|
> |w/o \<next_step\>|73.4|
> |w/o \<direct_answer\>|75.0|
> |w/o \<verify\>|72.8|
> |w/o \<refine\>|72.6|
> |All (Ours)|79.8|
>
> Table R3.
> |Models|Settings|MATH-500|GSM8K|SVAMP|Olympiad Bench|AMC-23|Avg.|
> |---|---|---|---|---|---|---|---|
> |Qwen3-4B-Base|RFTT (Origin)|83.4|96.1| 94.7| 48.1| 75.0| **79.46**|
> ||RFTT training w/o \<verify\>, \<refine\>|82.4|95.6|93.1|46.8|70.0|77.58|
>
> **[W3 & Q1] potential reward hacking problem.**
>
> Thanks for your insightful question! In practice, we combine outcome-based reward with supplemented by PRM signals, which aims to balance between both accuracy and quality of reasoning process. We further analysed the average process rewards (ranging from -0.5 to 0.5) of trajectories generated by models on the MATH-500 dataset. The results in Tab. R4 show that the average PRM scores of incorrect trajectories are much lower than those of correct trajectories, which may alleviate the problem of reward hacking.
>
> Table R4.
> | Models| Avg. PRM scores of correct samples | Avg. PRM scores of incorrect samples |
> |---|---|---|
> |Qwen3-4B-Base|0.38|-0.04|
> |Qwen2.5-7B-Instruct|0.34|0.02|
> |LLaMA-3.1-8B-Instruct|0.31|-0.05|

---

> ### Author Response · Authors · 2025-11-22
> **Response (2): Q3 to Q4**
>
> **[Q3] During functional tree search, what criteria are used to select the optimal reasoning branch?**
>
> Thanks for your question!
>
> (1) During SFT data construction, we choose a correct trajectory with the highest average process reward in the reasoning tree as stated in Section 3.2. The motivation is to ensure both the correctness of the final answer and the quality of reasoning processes for SFT data.
>
> (2) During RL training, we employ functional token-guided tree search but do not select optimal trajectories. The model learns which token combinations contribute to golden answers under current states through both positive and negative samples.
>
> **[Q4] How to get the dependencies between actions?**
>
> Thanks for the question! Motivated by rStar [1], the dependencies derived from analysis of human reasoning patterns yet semantically grounded, balancing flexibility with structural coherence. For example, \<clarify\> typically appears at the beginning of the problem, while \<output\> appears at the end to provide an expected answer. Empirically, we further investigate the reasoning behavior of Deepseek-R1 on MATH-500. Specifically, the sampled reasoning trajectories are broken down into discrete steps using predefined rules (e.g., newline delimiters). We then employ GPT-4.1 to map each step to a corresponding functional token. After analyzing the dependencies of each trajectory, we find 95.2% (476 of 500) transitions agree with the dependencies. We observe that transitions that do not satisfy the dependencies are like \<output\> -> \<next_step\>, \<subquestion\> -> \<output\>, \<subquestion\> -> \<refine\>, *etc.*. These invalid transitions do not conform to normal reasoning logic, which seems to be noise rather than genuine reasoning needs. Overall, the dependencies prune unproductive branches to improve search efficiency while preserving reasonable flexibility.
>
> ---
> [1] Mutual Reasoning Makes Smaller LLMs Stronger Problem-Solvers, ICLR 2025.

---

### Official Review · Reviewer_Wmaw · 2025-10-30

**Soundness:** 2
**Presentation:** 3
**Contribution:** 2
**Rating:** 4
**Confidence:** 5

**Summary:**

The paper proposes Reinforced Functional Token Tuning (RFTT), a two-phase framework to enhance LLM reasoning via functional token–guided Monte Carlo Tree Search. During the SFT phase, the model learns reasoning behaviors, such as <analyze>, <verify>, <refine>) from self-generated trajectories. In the RL phase, it explores diverse reasoning paths using these tokens as actions, optimized by process and outcome rewards. Experiments on math and general reasoning benchmarks show improved performance over GRPO and ReFT, suggesting functional tokens enable structured, self-improving reasoning in smaller LLMs.

**Strengths:**

1. The paper is clearly written and well organized, with a logical flow from motivation to methodology and experiments. Figures and tables effectively illustrate the framework and results.

2. The method is sound, with detailed descriptions of the two-phase (SFT + RL) pipeline, reward design, and MCTS procedure. All experiment settings are reasonable and detailed across different domain and model scale size.

**Weaknesses:**

1. The overall framework is somewhat similar to Satori (Feb 2025) and TreeRL (Jun 2025), as well as other recent RL-based reasoning frameworks that employ token-level actions or tree-structured exploration. It would be helpful to clarify how your method differs from these tree-RL-like approaches and to provide side-by-side results with these baselines [1, 2].

2. The current comparisons include only ReFT and GRPO, omitting strong recent baselines such as TreeRL and DAPO. Adding at least one of these baselines—or discussing the expected difference under a comparable compute budget—would make the evaluation more convincing [2, 3].

3. The paper attributes its improvement to high-entropy branching, but it remains unclear whether this approach provides a clear advantage over existing entropy-based RL methods, which often achieve similar goals without the added cost of increased training time. Since several recent studies have explicitly examined entropy mechanisms to enhance RL for reasoning LMs [4], a comparison with those methods would make the claim more solid and help clarify whether the added training cost is justified.


[1] Satori: Functional Token Guided Reinforcement Learning for Structured Reasoning, Feb 2025.
[2] TreeRL: Tree-Structured Reinforcement Learning for Multi-Step Reasoning, Jun 2025.
[3] DAPO: Diversity-Aware Policy Optimization for LLM Reasoning, May 2025.
[4] The Entropy Mechanism of Reinforcement Learning for Reasoning Language Models, May 2025.

**Questions:**

See Weaknesses

---

> ### Author Response · Authors · 2025-11-22
> **Response (1): W1 and W2**
>
> We sincerely appreciate the reviewer for finding that our paper is well organized and well motivated, with reasonable and detailed experiments. Below we address the main points raised in the review.
>
> **[W1 & W2] clarify how your method differs from these tree-RL-like approaches and to provide side-by-side results with these baselines.**
>
> Thanks for your suggestion!
>
> (1) Different from TreeRL [1]. TreeRL first samples M trajectories and then identifies N tokens that has high entropys. After that, the model continues to generate the rest responses from the positions of these tokens. As we discussed in our paper,  LLMs tend to generate homogeneous reasoning paths due to their inherent preference for syntactic patterns established during pre-training. For advanced reasoning patterns such as self-reflection and self-correction that are not learned in pre-training, it is still difficult to explore them simply by sampling from high-entropy tokens. What's more, TreeRL indeed adopts vocabulary-level actions to conduct tree search on LLMs, which has a larger search space and reduced search efficiency. For example, some meaningless punctuation marks, numbers, or symbols may have high entropy, branching and sampling in these positions may actually disrupt the integrity of the reasoning. In contrast, we employ token-guided tree search, which reduces the search space while allowing us to explore semantically different reasoning patterns.
>
> (2) Different from Satori [2]. Satori does not actually use tree search in RL framework. Satori introduces three reasoning actions and uses Restart and Explore (RAE) algorithm. RAE augments training data by randomly backtracking trajectories and inserting reflection tokens in their intermediate step. During RL, the model samples from both original problems and these synthetically intermediate states. Its exploration is static and manually designed rather than adaptive to training dynamics. Moreover, random insertion of reflection tokens at intermediate steps may be suboptimal. Conversely, RFTT conducts dynamic, token-guided tree search during rollout, sampling functional tokens based on the policy's probabilities and UCT exploration scores. This enables the model to automatically discover and reinforce optimal functional token combinations through reward signals, rather than relying on manual backtracking to restart exploration.
>
> (3) More baseline results and discussion. As suggested, we provide the results of TreeRL in Table R1 and DAPO [3] in Table R3, respectively. The results in Tab. R1 and Tab. R2 shows RFTT outperforms TreeRL with even fewer generated tokens. *The additional results of TreeRL have been updated in Table 1 of our revised manuscript.*
>
> Table R1. Comparison with TreeRL
> |Models|Settings|MATH-500|GSM8K|SVAMP|Olympiad Bench|AMC-23|Avg.|
> |-|-|-|-|-|-|-|-|
> |Qwen3-4B-Base|TreeRL|82.2|95.3|92.8|46.5|70.0|77.36|
> ||RFTT (Ours)|83.4|96.1|94.7|48.1|75.0|**79.46**|
> |Qwen-2.5-7B-Instruct |TreeRL|78.4|94.8|92.3|39.6|62.5|73.52|
> ||RFTT (Ours)|79.8|95.2|93.0|40.3|70.0|**75.66**|
> |Llama-3.1-8B-Instruct|TreeRL|57.8|92.6|86.1|26.7|47.5|62.14|
> ||RFTT (Ours)|60.2|91.9|87.5|29.8|55.0|**64.88**|
>
> Table R2. Token budget comparison with TreeRL on Qwen2.5-7B-Instruct
> |Method|Generated Token (k)|
> |-|-|
> |TreeRL|524.6 k / step|
> |RFTT (Ours)|216.8 k / step|
>
> Table R3. Comparison with DAPO
> |Models| Settings|MATH-500|GSM8K|SVAMP|Olympiad Bench|AMC-23|
> |-|-|-|-|-|-|-|
> |Qwen3-4B-Base|DAPO|79.4|93.5|90.1|44.7|70.0|
> ||RFTT (Ours)|83.4|96.1|94.7|48.1|75.0|
> ||RFTT + DAPO|85.0|96.4|95.2|49.8|77.5|
> |Llama-3.1-8B-Instruct|DAPO|56.4|90.1|84.5|25.3|27.5|
> ||RFTT (Ours)|60.2|91.9|87.5|29.8|55.0|
> ||RFTT + DAPO|62.4|92.7|88.2|30.5|57.5|
>
> Note that DAPO adds a diversity term $J_{Div} (\pi_\theta)$ to the original RL objective to facilitate exploration. While our method promotes exploration by improving the diversity of sampling. The two methods operate at different aspects but do not conflict with each other. The results in Table R3 show that RFTT consistently outperforms DAPO, while adding this diversity term to our loss functions can further improve the performance of RFTT.

---

> ### Author Response · Authors · 2025-11-22
> **Response (2): W3**
>
> **[W3] Whether this approach provides a clear advantage over existing entropy-based RL methods.**
>
> We thank the reviewers for this critical question. Similar to [W2] (3), RFTT promotes diversity at the sampling level for exploration, while Cui et al.[4] restricts gradient updates of tokens with excessively high covariance at the loss function level. Therefore, our method is fundamentally **orthogonal** to existing entropy-based RL methods: we operate at different levels but do not conflict with each other. The results in Table R4 show that RFTT significantly outperforms Clip-Cov, while adding these tricks to our loss function can further improve the performance of our method.
>
> Table R4. Comparison with entropy-based RL
> |Models| Settings|MATH-500|GSM8K|SVAMP|Olympiad Bench|AMC-23|
> |-|-|-|-|-|-|-|
> |Qwen3-4B-Base|Clip-Cov [4]|76.4|92.5|89.1|43.3|70.0|
> ||RFTT (Ours)|83.4|96.1|94.7|48.1|75.0|
> ||RFTT + Clip-Cov|83.8|95.7|95.4|48.7|75.0|
> |Llama-3.1-8B-Instruct|Clip-Cov   |54.6|89.3|84.4|25.0|27.5|
> ||RFTT (Ours)|60.2|91.9|87.5|29.8|55.0|
> ||RFTT + Clip-Cov|60.8|92.6|88.2|30.4|55.0|
>
> Once again, we sincerely appreciate the reviewer’s valuable feedback; it will substantially strengthen the paper’s positioning and empirical evidence.
>
> ---
> [1] TreeRL: LLM Reinforcement Learning with On-Policy Tree Search
>
> [2] Satori: Reinforcement Learning with Chain-of-Action-Thought Enhances LLM Reasoning via Autoregressive Search
>
> [3] Diversity-Aware Policy Optimization for Large Language Model Reasoning
>
> [4] The Entropy Mechanism of Reinforcement Learning for Reasoning Language Models

---

### Official Review · Reviewer_Hzqr · 2025-11-01

**Soundness:** 3
**Presentation:** 3
**Contribution:** 3
**Rating:** 6
**Confidence:** 4

**Summary:**

The paper proposes **Reinforced Functional Token Tuning (RFTT)**, a two-phase framework to endow LLMs with learn-to-reason abilities by internalizing *functional tokens*. Phase I performs SFT on training data generated by a functional-prompt-guided MCTS that stitches together correct and incorrect branches via cross verification and branch merging, yielding structured, token-annotated reasoning paths. Phase II replaces prompt guidance with token-guided exploration and optimizes with Reinforce++ under a KL penalty to a reference SFT model, optionally using a PRM for step-level rewards. On math benchmarks (MATH-500, GSM8K, SVAMP, OlympiadBench, AMC), RFTT improves small models (≤10B) markedly over CoT, self-consistency, ReFT, and GRPO, and shows monotonic gains with more inference-time rollouts. Ablations indicate benefits from SFT warmup, MCTS, and PRM, while token masking highlights the importance of `<verify>` and `<refine>`. The authors also report training dynamics, cost trade-offs, and limitations. Overall, the contribution is a principled way to compress a vast vocabulary-level action space into a compact, semantically meaningful policy over functional tokens, enabling more effective exploration and reinforcement of reasoning behaviors.

**Strengths:**

The work is original in elevating “functional prompts” (e.g., from rStar-style search) into learnable tokens that become part of the model’s action space and are optimized end-to-end. This shift from external, prompt-level constraints to internalized, token-level policies is a clean conceptual advance that reduces search branching at the word level while preserving diversity (§3.1–3.3, pp.3–5).

The qualityof the methodology is supported by a careful algorithmic design—cross-verification and branch-merging produce structured SFT targets (§3.2, p.4), token-guided MCTS with UCT balances exploration/exploitation (Eqns. (1),(5)–(8), pp.4,23), and PRM-based process rewards address long-chain credit assignment (§4.4, Table 3 on p.8). Empirically, the paper demonstrates clear, consistent gains across multiple backbones (Qwen-3-4B, Qwen-2.5-7B, LLaMA-3.1-8B) and datasets, with strong improvements versus ReFT and GRPO and favorable scaling with additional rollouts (Table 1, Figure 2, pp.6–7).

The exposition is clear, with a unifying diagram (Figure 1, p.3), explicit token definitions and prompts (Appendix C, pp.18–19), and transparent training/eval details (§4.2, p.6). RFTT offers a general recipe to couple MCTS and RL via a compact, behavior-level action space, pushing small-model reasoning toward larger-model performance, and the ablations convincingly attribute where the gains come from (Table 3, p.8; Table 4 and Figure 3, p.8).

**Weaknesses:**

1. The OOD benchmarks are still somewhat in-distribution, as they are essentially math reasoning problems. It would be interesting to test models on other reasoning benchmarks, like FOLIO (logical reasoning), CruxEval (code reasoning), TableBench (table reasoning), etc.

2. Baseline methods are not comprehensive. Performance comparisons with more recent tuning based methods (e.g. rStar-math, Satori, MCTS-DPO) and non-tuning based methods (e.g. rStar, RAP) would bring more insights into the strength of this work.

3. Scalability test of the method is missing. Experiements on smaller/larger models would also be interesting.

4. Showing these heuristic designs of reasoning actions also work on other reasoning domains, like code generation, for synthesizing training data, would also strengthen the work.

**Questions:**

1. Could you elaborate the motivation of using PRM? How does it compare with ORM?

---

> ### Author Response · Authors · 2025-11-22
> **Repsonse to Weakness and Question**
>
> We greatly appreciate the reviewer for your insightful comments and positive support.
>
> **[W1] Test models on other reasoning benchmarks**
>
> Thanks for your valuable suggestion! To demonstrate broader OOD generalization, we provided the results on benchmarks FOLIO (logical reasoning), CruxEval (code reasoning), and TableBench (table reasoning) in Table R1. The results show that RFTT can achieve promising performance across a wide range of reasoning tasks, despite being trained solely on mathematical data. *The additional results have been updated in Tab.2 of our revised manuscript.*
>
> Table R1
> |Models|Settings|FOLIO|TableBench|CRUXEval|
> |-|-|-|-|-|
> |Qwen-2.5-7B-Instruct |Fewshot-CoT|72.9|43.6|**58.3**|
> ||RFTT (Ours)|**73.9**|**44.4**|57.5|
> |Llama-3.1-8B-Instruct|Fewshot-CoT|64.5|32.8|38.6|
> ||RFTT (Ours)|**65.6**|**34.2**|**39.5**|
>
> **[W2] Baseline methods are not comprehensive**
>
> Thanks for your suggestion. We further expand our comparisons to training-based methods like MCTS-DPO, TreeRL in Table R2 and non-tuning based methods like rStar, LLaMA-Berry in Table R4. The results in Tab. R2, R3 show RFTT outperforms TreeRL with even fewer generated tokens. *The additional results have been updated in Tab.1 and Tab.4 of our revised manuscript.*
>
> Table R2. Comparison with training-based methods
> |Models|Settings|MATH-500|GSM8K|SVAMP|Olympiad Bench|AMC|Avg.|
> |-|-|-|-|-|-|-|-|
> |Qwen3-4B-Base|TreeRL|82.2|95.3|92.8|46.5|70.0|77.36|
> ||MCTS-DPO|73.8|90.2|86.9|39.7|60.0|70.12|
> ||RFTT (Ours)|83.4|96.1|94.7|48.1|75.0|**79.46**|
> |Qwen-2.5-7B-Instruct|TreeRL|78.4|94.8|92.3|39.6|62.5|73.52|
> ||MCTS-DPO|74.6|92.5 |89.8 |37.2 |55.0 |69.82|
> ||RFTT (Ours)|79.8|95.2|93.0|40.3|70.0|**75.66**|
> |Llama-3.1-8B-Instruct|TreeRL|57.8|92.6|86.1|26.7|47.5|62.14|
> ||MCTS-DPO|53.6 |87.4|82.7|23.1|25.0|54.36|
> ||RFTT (Ours)|60.2|91.9|87.5|29.8|55.0|**64.88**|
>
> Table R3. Token budget comparison with TreeRL on Qwen2.5-7B-Instruct
> |Method|Generated Token (k)|
> |-|-|
> |TreeRL|524.6 k/step|
> |RFTT (Ours)|216.8 k/step|
>
> Table R4. Comparison with non-tuning-based methods on Llama3.1-8B-Instruct
> |Dataset|Method|Accuracy|Time per Rollout (s)|
> |-|-|-|-|
> |**GSM8K**|rStar|92.1%|276|
> ||LLaMA-Berry|94.9%|339|
> ||**Ours**|**95.2%**|**81**|
> |**MATH-500**|rStar|61.0%|526|
> ||LLaMA-Berry|69.4%|674|
> ||**Ours**|**72.0%**|**131**|
>
> The model trained by RFTT can autonomously select the next functional token and the corresponding step based on a partially generated reasoning trajectory, thus enabling a more efficient exploration of the reasoning space without frequently interrupting the generation process in MCTS simulation stage. The results in Tab. R4 show RFTT outperforms rStar while maintaining search efficiency. Due to the time limit, we will supplement the results of RAP as soon as possible.
>
> **[W3] Miss scalability test of the method**
>
> Thanks for the helpful suggestion. In our original paper, we provide model sizes of 3B, 4B, 7B and 8B. As suggested, we provide additional results on Qwen3-14B-Base to further validate the effectiveness of RFTT when model parameter scaling. The results in Tab. R5 shows meaningful improvements, which demonstrate the effectiveness of our approach. *The additional results have been updated in Appendix B (Page 19) of our revised manuscript.*
>
> Table R5
> |Models|Settings|MATH-500|GSM8K|SVAMP|Olympiad Bench|AMC-23|Avg.|
> |-|-|-|-|-|-|-|-|
> |Qwen3-14B-Base|Few-shot CoT|75.8|92.5| 82.1| 40.5| 67.5| 71.70|
> ||ReFT|85.8|95.7|92.5|49.8|80.0|80.76|
> ||RFTT(Ours)|89.0|97.2|95.3|53.0|85.0| **83.90**|
>
> **[W4] Showing the designs of reasoning actions also work on other reasoning domains**
>
> As suggested, we have additionally conducted experiments using training data curated from non-mathematical domains as shown in Tab. R6. The results in Tab. R7 show broader improvements when RFTT is exposed to more diverse reasoning data, indicating strong potential for cross-domain generalization. *The additional results have been updated in Appendix B (Page 20) of our revised manuscript.*
>
> Table R6
> |Source|MMLU-Pro (health)|MMLU-Pro (biology)|MMLU-Pro (physics)|MMLU-Pro (chemistry)|MMLU-Pro (law)|HumanEval|
> |-|-|-|-|-|-|-|
> |**# Samples**|267|144|328|135|126|160|
>
> Table R7
> |Settings| MMLU (STEM)|MMLU (Professional Medicine)|MMLU (Professional Law)|GPQA| CommonsenseQA|MBPP (Code)|
> |-|-|-|-|-|-|-|
> |Few-shot CoT|81.1|77.6|50.4|36.4|81.3|79.2|
> |RFTT (Ours)|**86.9**|**83.1**|**55.8**|**38.8**|**85.2**|**83.8**|
>
> **[Q1] The motivation of using PRM and compare with ORM**
>
> ORM provides a sparse, outcome-only signal (correct/incorrect). In long-horizon reasoning, sparse rewards cause high variance and poor credit assignment for each token. PRM provides fine-grained rewards for intermediate reasoning steps, which teaches the model to learn the quality at each generated step. However, PRM may introduce domain dependence and noise. Therefore, we still integrated outcome-based reward when using PRM score, which achieves a balance between both accuracy and quality of reasoning process.

---

> ### Author Response · Authors · 2025-11-28
> **Response: Additional Results of RAP**
>
> Dear Reviewer Hzqr,
>
> We further provide the results of non-tuning-based method RAP in Table R8. The results show that our method can achieve higher performance.
>
> Table R8. Comparison with non-tuning-based methods on Llama3.1-8B-Instruct
> |Dataset|Method|Performance|
> |-|-|-|
> |**GSM8K**|rStar|92.1%|
> ||RAP|87.4%|
> ||**Ours**|**95.2%**|
> |**MATH-500**|rStar|61.0%|
> ||RAP|55.8%|
> ||**Ours**|**72.0%**|
>
> We are very grateful for your thoughtful and detailed feedback on our manuscript. We believe that incorporating your suggestions has substantially improved the quality of the manuscript.

---

### Official Review · Reviewer_b1q7 · 2025-11-02

**Soundness:** 2
**Presentation:** 2
**Contribution:** 2
**Rating:** 4
**Confidence:** 3

**Summary:**

This paper proposes to train LLMs to reason by introducing new special tokens (i.e., functional tokens in this paper) and combining it with tree search. The proposed method warm-up the model with SFT data that teach the model how to follow the functional tokens. The second stage trains the model to generate diverse reasoning paths with MCTS. The results show that the proposed method (RFTT) improves the performance over RL fine-tuning on math reasoning tasks.

**Strengths:**

The idea of combining tree search and RL during training is a promising idea since it may improve data efficiency significantly.

**Weaknesses:**

- Writing is not clear. See my questions.
- Training model with Equation 4 with the data collected by MCTS sampling (Equation 1) is biased. The PPO's objective needs to sample states and actions from the policy pi_old, but MCTS sampling draws actions based on UCT (Equation 1), which won't yield an accurate expectation on the clipped objective L_RL
- The ablation study (Table 3) shows the setting of ablating SFT-warmup, MCTS, and PRM, but it's unclear that if MCTS and PRM are required. One possibility is that RFT + SFT-warmup would yield similar performance.

**Questions:**

- Line 64: What does human-like reasoning tree mean?
e
- Line 68: What does "prompt-guided" mean?
- Line 69: Why the exploration is diverse needs to be explained.
- Line 172: How do you tell the intersection?
- Line 166-Line 180: The notations are very hard to parse.
- Line 210: Unclear what ANS predicts.
- Line 348: It's not rigourous to use "siginifcantly improve" when there are no confidence interval on the results.
- Line 352: What is self-consistency sampling?
- Line 363: I'm confused. I thought you need to add functional tokens to the dictionary also in RL/SFT training, but it reads like you only add them to the dictionary while doing tree search?
- Line 365: Without reward model or any other reward signal, how do you perform MCTS?
- Line 204: UCT score's definition is missing. Though it was a classic notion in MCTS, it's better to clarify the definition in the context of this paper.
- Line 262: What is "MATH dataset"? Needs reference.

---

> ### Author Response · Authors · 2025-11-22
> **Response (1): Weakness**
>
> Thanks for your detailed and constructive feedback. Below, we first briefly clarify our pipeline and then provide point-by-point responses to your concerns and questions.
>
> Some studies (Shah et al., 2025 [1]; Yue et al., 2025 [2]) suggest that 1) RL may not introduce fundamentally new reasoning capabilities, but rather increase the likelihood of sampling efficiency. 2) Moreover, RL enhances sampling efficiency at the cost of reducing reasoning diversity, limiting the exploration capability.
>
> To address challenge (1), we construct self-generated training data annotated with several functional tokens to equip initial models with superior reasoning behaviors via SFT Warmup. Specifically, we introduce several human-like reasoning behaviors as functional tokens (e.g., \<verify\>, \<refine\>) and perform MCTS using corresponding functional prompts presented in Appendix C. Then, we synthesize training trajectories annotated with functional tokens by cross verification and branch merging (Section 3.2) for SFT. During SFT Warmup, these learnable functional tokens are embedded into model vocabulary to facilitate initial reasoning capability. To address challenge (2), the RL phase employs functional tokens that learned in SFT Warmup to conduct tree search. Specifically, the model transitions from prompt-guided to token-guided reasoning by sampling functional tokens to autonomously explore diverse reasoning nodes.
>
> Overall, we employ functional prompts as the action space for expanding tree nodes during the preliminary SFT warmup phase, while transitioning to token-guided tree traversal during RL sampling. The SFT phase bootstraps reasoning through self-generated training data annotated with functional tokens. After being trained with functional tokens, the policy model $\pi_\theta$ can explicitly and dynamically select functional tokens and further generate the specific reasoning steps that align with the semantics of these tokens.
>
> ---
>
> **[W1] Training model with Eq. (4) with the data collected by MCTS sampling is biased.**
>
> Thanks for the insightful question! As we summarized before, after embedding these functional tokens into the model vocabulary in SFT stage, the policy model $\pi_\theta$ can dynamically sample them and further generate the specific reasoning steps that align with the semantics of functional tokens. We are mindful of potential distribution mismatch, but we believe the bias may not be severe in our setup. First, our action space consists of eight functional tokens, which is much smaller than the vocabulary-level action space. The distribution differences between policy $\pi_{old}$ and UCT selection are relatively controllable in this low-dimensional space, reducing the overall variance and bias of importance sampling ratios. Second, the clipping mechanism is inherently more robust to mildly off-policy drift by truncating ratios exceeding the clipping interval. We also normalize advantages and apply a KL regularizer, which further constrains policy updates by the presence of mild distributional shifts. Empirically, the impact of distributional mismatch appears small in our setting: the RFTT w/o PRM (MCTS sampling and rule-based RL after SFT) setting is stable and consistently outperforms the RFTT w/o MCTS (vanilla sampling and rule-based RL after SFT) settings. For your convenience, we present relevant results from Table 3 in our manuscript in the following table again.
>
> |Models|Setting|MATH-500 |GSM8K |SVAMP|Olympiad Bench|AMC|Avg.|
> |-|-|-|-|-|-|-|-|
> |Qwen2.5-7B-Instruct| RFTT w/o MCTS |75.2|93.8| 91.2| 39.0| 62.5| 72.34|
> ||RFTT w/o PRM|77.2|94.1|92.4|38.4|67.5|**73.92**|
> |LLaMA-3.1-8B-Instruct |RFTT w/o MCTS|56.2|91.5| 84.0| 29.1| 42.5| 60.66|
> ||RFTT w/o PRM|57.4|92.2|86.3|28.6|50.0|**62.90**|
>
> **[W2] it's unclear that if MCTS and PRM are required.**
>
> Sorry for the confusion. As stated in the first paragraph of Section 4.4 in our original manuscript, the ablation setting RFTT w/o MCTS employs vanilla sampling and rule-based RL after SFT Warmup, which is the same meaning with RFT + SFT-warmup you mentioned. For your convenience, we include again here the results from Table 3 in our manuscript as Table R1. The results in Table R1 of our manuscript show that our method has an improvement of over 3% compared to the RFT + SFT-warmup settings.
>
> Table R1.
> | Models| Setting| MATH-500 | GSM8K | SVAMP | Olympiad Bench| AMC| Avg. |
> | - | - | - | - | - | - | - | - |
> |Qwen2.5-7B-Instruct| RFTT w/o MCTS |75.2|93.8| 91.2| 39.0| 62.5| 72.34|
> ||RFTT (Ours)|79.8|95.2| 93.0| 40.3| 70.0| **75.66**|
> |LLaMA-3.1-8B-Instruct |RFTT w/o MCTS|56.2|91.5| 84.0| 29.1| 42.5| 60.66|
> ||RFTT (Ours)|60.2|91.9|87.5|29.8|55.0|**64.88**|

---

> ### Author Response · Authors · 2025-11-22
> **Response (2): Q1 to Q6**
>
> **[Q1] What does human-like reasoning tree mean?**
>
> It refers to a tree structure where nodes represent distinct human-like reasoning behaviors (*e.g.*, clarify, analyze). Unlike standard trees that expand arbitrary next steps, our tree explicitly models human cognitive patterns through functional tokens, creating branches that model how humans are involved in complex reasoning.
>
> **[Q2] What does "prompt-guided" mean?**
>
> In SFT training data construction, we use explicit functional prompts (Appendix C) to induce the model to produce the corresponding type of step. These prompts act as external constraints to collect structured reasoning steps annotated with functional tokens, as shown in Appendix D. After SFT Warmup, the model learns to reason using these functional tokens, where the model directly samples them from its vocabulary and generates a semantically consistent step without relying on external prompts. *These clarifications have been updated in Section 3.1 of the revised manuscript.*
>
> **[Q3] Why the exploration is diverse needs to be explained.**
>
> LLMs tend to generate homogeneous reasoning paths due to their inherent preference for syntactic patterns established during pre-training. In our work, the policy samples distinct reasoning actions (clarify, verify, refine, etc.), each leading to semantically different reasoning directions. As shown in Fig. 5, middle, the entropy during early training steps exhibits an increasing trend, suggesting a diverse exploration at the early stage. *These clarifications are in Section 2 (line 104) and Section 3.2 (line 155) of the original manuscript.*
>
> **[Q4] How do you tell the intersection?**
>
> For two reasoning trajectories extracted from reasoning trees, if they have the same ancestor nodes (excluding the root node), we say the two trajectories intersect. For example, trajectories $\tau_1 = \[ s_{0}, s_{2}, s_{16}, s_{22}, s_{40}, s_{56} \]$ and $\tau_2 = \[ s_0, s_2, s_{16}, s_{22}, s_{48}, s_{59} \]$ intersect at node $s_{22}$.
>
> **[Q5] The notations are very hard to parse.**
>
> Sorry for the confusion. Our intention was to use rigorous mathematical expressions to explain this process. We now describe this part in more concise natural language. Technically, we choose a correct trajectory $ \tau_c $ with the highest average process rewards and an incorrect trajectory $\tau_w$ that intersects with $ \tau_c $ with the lowest average process rewards. We denote the last intersection node as $s$. After that, we prompt the model to reflect where's wrong with $ \tau_w $ after reasoning step $s$ compared to $\tau_c$. Thus we can yield a verification node $s_v$. Finally, we use $ s_v $ to connect the incorrect and correct trajectories, forming a complex reasoning structure with self-reflection and self-correction. *Additionally, we further add Fig.2 in our revised manuscript to help understand cross-verification and branch-merging more intuitively.*
>
> **[Q6] Unclear what ANS predicts.**
>
> $\text{ANS}(\cdot)$ is a **rule-based answer extractor** that extracts the final answer from a reasoning step. For mathematical problems, it parses the expression or numeric answer wrapped in `\\boxed{}`. If $\text{ANS}(s_t)$ is null, it represents $s_t$ is the reasoning process that does not contain a final answer. We'll define this explicitly when first introduced. *These clarifications have been updated in line 222 of the revised manuscript.*

---

> ### Author Response · Authors · 2025-11-22
> **Response (3): Q7 to Q12**
>
> **[Q7] It's not rigourous to use "significantly improve" when there are no confidence interval on the results.**
>
> To achieve reproducible results, we evaluate under a sampling temperature of 0.7 under a fixed seed 42 across all datasets. As suggested, we report the metrics with confidence interval over 5 independent runs with random seeds in Tab. R2. We find our reported results fall within confidence intervals and are almost identical to the mean value.
>
> Table R2.
> |Models|Settings|MATH-500|GSM8K|SVAMP|Olympiad Bench|AMC-23|
> |--|--|--|--|--|--|--|
> |Qwen-3-4B-Base|Zero-shot CoT|58.4 ± 0.8|80.5 ± 0.7|78.9 ± 0.5|31.2 ± 0.4|49.5 ± 2.9|
> ||Few-shot CoT|64.7 ± 0.6|87.5 ± 0.5|80.5 ± 0.4|35.5 ± 0.5|57.0 ± 2.9|
> ||CoT+SC@4|67.1 ± 0.7|87.9 ± 0.7|81.4 ± 0.6|36.1 ± 0.5|54.0 ± 2.5|
> ||ReFT|79.0 ± 0.5|94.3 ± 0.4|91.6 ± 0.3|45.5 ± 0.6|68.5 ± 3.0|
> ||GRPO|75.3 ± 0.8|92.5 ± 0.6|88.5 ± 0.2|42.9 ± 0.3|66.5 ± 2.5|
> ||**SFT Warmup (Ours)**|68.2 ± 0.9|89.6 ± 0.8|85.3 ± 0.6|38.8 ± 0.4|61.5 ± 2.5|
> ||**RFTT (Ours)**|83.5 ± 0.7|96.4 ± 0.5|94.8 ± 0.3|48.2 ± 0.2|74.5 ± 2.9|
> |Qwen-2.5-7B-Instruct|Zero-shot CoT|72.6 ± 1.1|91.3 ± 0.6|88.3 ± 0.4|35.6 ± 0.9|48.0 ± 4.8|
> ||Few-shot CoT|76.0 ± 0.9|91.7 ± 0.8|90.3 ± 0.5|37.0 ± 0.8|53.4 ± 3.2|
> ||CoT+SC@4|76.6 ± 1.0|92.2 ± 0.6|90.8 ± 0.4|38.1 ± 0.6|52.5 ± 4.7|
> ||ReFT|75.9 ± 1.2|94.3 ± 0.7|89.6 ± 0.4|38.4 ± 0.5|61.0 ± 4.6|
> ||GRPO|76.2 ± 1.2|93.5 ± 0.6|90.4 ± 0.3|36.5 ± 0.7|55.5 ± 3.3|
> ||**SFT Warmup (Ours)**|73.5 ± 0.9|92.1 ± 0.9|87.7 ± 0.6 |36.7 ± 0.8|61.5 ± 4.4|
> ||**RFTT (Ours)**|80.1 ± 0.8|95.3 ± 0.5|93.1 ± 0.2|40.6 ± 0.5|70.5 ± 4.3|
> |LLaMA-3.1-8B-Instruct|Zero-shot CoT|50.4 ± 1.3|84.5 ± 1.0|78.3 ± 0.5|18.1 ± 1.2|17.9 ± 5.2|
> ||Few-shot CoT|52.8 ± 1.0|85.4 ± 0.8|81.7 ± 0.4|19.7 ± 0.9|22.0 ± 4.3|
> ||CoT+SC@4|52.3 ± 1.2|86.2 ± 0.7|85.5 ± 0.3|19.9 ± 0.8|26.5 ± 4.3|
> ||ReFT|55.2 ± 0.8|90.6 ± 0.5|84.4 ± 0.2|24.9 ± 0.6|43.5 ± 4.1|
> ||GRPO|54.4 ± 1.1|88.0 ± 0.6|83.6 ± 0.3|23.8 ± 0.7|23.5 ± 4.6|
> ||**SFT Warmup (Ours)**|53.1 ± 0.9|86.7 ± 0.9|82.6 ± 0.5|24.3 ± 0.9|40.5 ± 3.3|
> ||**RFTT (Ours)**|60.3 ± 0.7|92.1 ± 0.4|87.6 ± 0.3|29.6 ± 0.5|57.0 ± 3.7|
>
> **[Q8] What is self-consistency sampling?**
>
> Sorry for the confusion. Self-consistency refers to the baseline CoT+SC proposed by Wang et al [3], which samples multiple trajectories and selects the answer with the highest frequency. *These clarifications have been updated in Section 4.3 of the revised manuscript.*
>
> **[Q9] I thought you need to add functional tokens to the dictionary also in RL/SFT training, but it reads like you only add them to the dictionary while doing tree search?**
>
> Sorry for the confusion. Functional tokens are added to the model vocabulary in SFT and RL stages. The SFT phase learns how to use these tokens for structural reasoning. During RL, the model continues to use these learned functional tokens to conduct MCTS. So we call this "token-guided tree search", which means the model samples these *already-learned* functional tokens directly from its vocabulary and generates semantically consistent step, without relying on external prompts. Therefore, functional tokens exist in both SFT and RL phase. We employ them for token-guided MCTS in RL phase, but not adding them to the dictionary only when doing tree search. For example, the settings RFTT w/o MCTS employ random sampling and rule-based RL after SFT Warmup, which does not utilize functional tokens for tree search, but the functional tokens still appear in the sampling trajectories.
>
> **[Q10] Without reward model or any other reward signal, how do you perform MCTS?** Sorry for the confusion. Following the settings of rStar [4], we use the likelihood (confidence) of self-consistency majority voting as the reward value. Specifically, we sample multiple solutions for the terminal node in the simulation phase, and use the maximum likelihood among all possible answers as the Q value. *These clarifications have been updated in Section 4.3 of the revised manuscript.*
>
> **[Q11] UCT score's definition is missing.** Thanks for the suggestion! We present this definition in Appendix E of our original manuscript for the sake of brevity. Specifically, $UCT(s) = \frac{Q(s)}{N(s)} + c\cdot\sqrt{\frac{\ln \tilde{N}(s)}{N(s)}}$, where $N(s_t)$ denotes the number of times step $s_t$ has been visited in previous iterations, and $\tilde{N}(s_t)$ is the visiting count of the parent node of $s_t$, $Q(s_t)$ is the estimated reward value and will be updated through back-propagation, and $c$ is a constant that balances exploitation and exploration. We agree that it's better to clarify it *in Section 3.3.2 (Page 4) of our revised manuscript.*
>
> **[Q12] What is "MATH dataset"? Needs reference.** Thanks! *We added the relevant citation Hendrycks et al. [5] in our revised manuscript.*

---

> ### Author Response · Authors · 2025-11-22
> **Response (4): References**
>
> We apologize for any inconvenience these multiple replies may cause and sincerely thank the reviewer for the thoughtful comments that help to improve the quality of our manuscript. We would like to know if we have clearly clarified these questions in this version. Looking forward to your feedback!
>
> ---
>
> [1] Rethinking reflection in pre-training. Shah et al.
>
> [2] Does reinforcement learning really incentivize reasoning capacity in llms beyond the base model? Yue et al., NeurIPS 2025.
>
> [3] Self-Consistency Improves Chain of Thought Reasoning in Language Models. Wang et al., ICLR 2023.
>
> [4] Mutual Reasoning Makes Smaller LLMs Stronger Problem-Solvers. Qi et al., ICLR 2025.
>
> [5] Measuring mathematical problem solving with the MATH dataset. Hendrycks et al., NeurIPS 2021.

---

### Meta-Review · Area_Chair_3xqr · 2026-01-10

**Summary:**

This paper proposes to incorporate capability of monte carlo tree search for empowering large language model’s reasoning capability. To this end, warm-up supervised fine-tuning is first conducted to learn functional tokens from self-generated training data with these tokens. Then, reinforcement learning is further conducted to enable autonomous diverse exploration of reasoning paths. The reviewers raised several concerns in their reviews, stemming from presentation issues and insufficient experiments (e.g., additional baselines, OOD evaluations, larger models). The authors provided comprehensive rebuttal with extensive additional experiments, but no reviewer responded to this.

**Reviewer Concerns:**

The reviewers' initial concerns and rebuttal by the authors can be summarized as below:

b1q7  (4, no response)

**[W1] Unclear writing.** The authors clarified in detail and provided additional results of standard deviation.

**[W2] Biased data collection due to UCT.** The authors clarified that the bias is not severe as the new action space is much smaller and inherent mechanisms such as clipping, normalization, and KL regularization are applied. Also, they remarked on the experimental results that support this claim.

**[W3] Absence of key ablation study.** The authors clarified that the pointed ablation study is already included, but it was not presented well due to some confusing parts.

Hzqr (6, no response)

**[W1, W4] Extension to challenging OOD benchmarks and domain.** The OOD evaluation on three additional reasoning benchmarks were presented. Also, training and evaluation on non-mathematical domain were also conducted.

**[W2] Additional baselines.** The authors provided new comparison results with two training-based RL baselines (TreeRL, MCTS-DPO) and two non-training methods (rStart, LLaMA-Berry).

**[W3] Generalization with smaller/larger models.** The experimental results on the larger model (Qwen3-14B) were provided in the rebuttal.

Wmaw (4, no response)

**[W1] Clarification compared to other token-level actions or tree-structured RL methods.** Detailed illustration of how RFTT is different with mentioned baselines is provided.

**[W2] Comparison with stronger baselines such as TreeRL and DAPO.** The authors conducted new experiments comparing TreeRL and DAPO to RFTT, and demonstrated clear advantages of RFTT in both accuracy and efficiency.

**[W3] Comparison with other entropy-based RL methods.** The authors provided a comparison with Clip-Conv baseline and also showed that RFTT is compatible with this.

xic9 (6, no response)

**[W1] Computational cost due to tree search and scalability.** The increase of computational cost is clarified and the experimental results on the larger model (Qwen3-14B) were provided.

**[W2] Limited flexibility and human bias due to pre-defined functional tokens.** The authors clarified the effectiveness of pre-defined functional tokens more concretely.

**[W3] Potential reward hacking.** The authors provided empirical evidence showing the reward hacking has merely not occurred.

**Reviewer Scores:**

Initially, the reviewers' scores were split (original: 4,6,4,6). Despite the authors providing the extensive rebuttal, no reviewers responded to this. After carefully reading all the reviews and rebuttals by the authors, AC believes that most of the concerns are now resolved and expect that all reviewers would have increased their scores to (expected: 6,6,6,6).

---

### Decision · Program_Chairs · 2026-01-26

Accept (Poster)